# MixPrompt: Efficient Mixed Prompting for Multimodal Semantic Segmentation

**Zhiwei Hao**[1][*] **Zhongyu Xiao**[1][*] **Jianyuan Guo**[2][†] **Li Shen**[3]**, Yong Luo**[4]**,**
**Han Hu**[1][†] **Dan Zeng**[5]

[1]School of information and Electronics, Beijing Institute of Technology.
[2]Department of Computer Science, City University of Hong Kong.
[3]School of Cyber Science and Technology, Sun Yat-sen University.
[4]School of Computer Science, Wuhan University.
[5]School of Communication and Information Engineering, Shanghai University.

{haozhw, zhongyu.xiao, hhu}@bit.edu.cn, jianyguo@cityu.edu.hk,
mathshenli@gmail.com, luoyong@whu.edu.cn, dzeng@shu.edu.cn

## Abstract

Recent advances in multimodal semantic segmentation show that incorporating auxiliary inputs—such as depth or thermal images—can significantly improve performance over single-modality (RGB-only) approaches. However, most existing solutions rely on parallel backbone networks and complex fusion modules, greatly increasing model size and computational demands. Inspired by prompt tuning in large language models, we introduce **MixPrompt**: a prompting-based framework that integrates auxiliary modalities into a pretrained RGB segmentation model without modifying its architecture. MixPrompt uses a lightweight prompting module to extract and fuse information from auxiliary inputs into the main RGB backbone. This module is initialized using the early layers of a pretrained RGB feature extractor, ensuring a strong starting point. At each backbone layer, MixPrompt aligns RGB and auxiliary features in multiple low-rank subspaces, maximizing information use with minimal parameter overhead. An information mixing scheme enables cross-subspace interaction for further performance gains. During training, only the prompting module and segmentation head are updated, keeping the RGB backbone frozen for parameter efficiency. Experiments across NYU Depth V2, SUN-RGBD, MFNet, and DELIVER datasets show that Mix-Prompt achieves improvements of 4.3, 1.1, 0.4, and 1.1 mIoU, respectively, over two-branch baselines, while using nearly half the parameters. MixPrompt also outperforms recent prompting-based methods under similar compute budgets.The code is available at https://github.com/xiaoshideta/MixPrompt.

## 1 Introduction

Semantic segmentation assigns a label to each pixel in an image and is a core task in computer vision. Progress in this area has largely been driven by large-scale RGB datasets such as Cityscapes [1] and ADE20K [2]. However, models trained on RGB data alone often struggle in challenging environments, such as low-light or poor weather, where visual cues are weak or missing. To address these limitations, multimodal segmentation approaches integrate data from additional sensors [3, 4, 5, 6, 7, 8, 9, 10, 11, 12, 13]. For example, RGB-D segmentation pairs RGB images with depth information, improving

---

[*]These authors contributed equally to this work.
[†]Corresponding to Han Hu and Jianyuan Guo.

39th Conference on Neural Information Processing Systems (NeurIPS 2025).

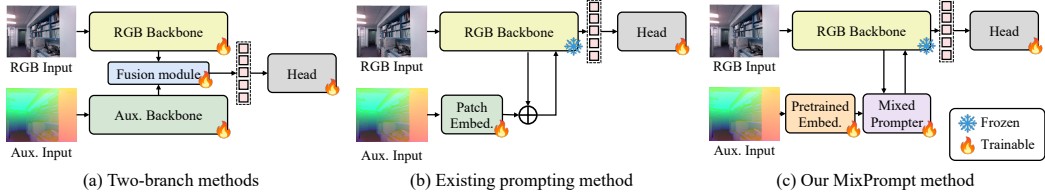

(a) Two-branch methods    (b) Existing prompting method    (c) Our MixPrompt method

Figure 1: Comparison between (a) conventional multimodal semantic segmentation framework, (b) prompting-based framework, and (c) our mixed prompting framework. Our framework excels by offering superior parameter efficiency while providing a more effective prompting module design.

object separation and spatial understanding [3, 4, 5, 6, 7, 8, 9]. As a result, multimodal methods have surpassed RGB-only models in both accuracy and robustness.

Despite these benefits, two main challenges remain for multimodal segmentation. **First**, model size increases significantly, adding more modalities usually means extra backbone branches and fusion modules [14], which raise computational costs and make deployment difficult on resource-limited platforms. **Second**, multimodal datasets are often much smaller than RGB ones, as collecting aligned data from auxiliary sensors is expensive and labor-intensive. For instance, NYU Depth V2 [15] contains only 1,449 RGB-D pairs, while ADE20K has over 25,000 RGB images. This data scarcity limits both training and generalization.

Recognizing these challenges, we draw inspiration from the vision-language domain [16, 17, 18], where prompt tuning allows a pretrained language model to process visual inputs by introducing visual features as prompts. This method efficiently adapts large models to new tasks or modalities with minimal data and parameters. We extend this idea to semantic segmentation by using a pretrained RGB model as the base, and introducing auxiliary modalities as prompts. Prior work sharing the similar idea [14] simply embeds data from the additional modality directly into the feature space and integrates it with the RGB features through simple addition, treating the modalities as basic addends rather than leveraging their complementary strengths. This method may not fully exploit the potential of the auxiliary modality, potentially leading to suboptimal performance.

To bridge this gap, we propose **MixPrompt**, a mixed prompting framework for efficient multimodal semantic segmentation. MixPrompt embeds auxiliary data using the initial layers of a pretrained RGB extractor, then aligns and fuses features from both modalities at multiple subspaces throughout the backbone. This approach maximizes information transfer with minimal parameter increase. As shown in Figure 1, MixPrompt achieves the efficiency of prompting-based methods while delivering stronger performance. We evaluate MixPrompt on four benchmark datasets: NYU Depth V2 [15], SUN-RGBD [19], MFNet [10], and DELIVER [8]. MixPrompt consistently outperforms strong two-branch and prompting-based baselines, achieving higher accuracy with fewer parameters.

The contributions of our paper can be summarized as follows:

- We introduce MixPrompt, an efficient prompting framework that integrates auxiliary modalities into a pretrained RGB segmentation model, ensuring both data and parameter efficiency.
- We show that reusing early layers of a pretrained RGB backbone is an effective embedding strategy for auxiliary modalities.
- We propose a multi-subspace alignment and prompting strategy that fully leverages auxiliary information, resulting in improved performance over existing baselines.

## 2 Related works

### 2.1 Semantic segmentation

Semantic segmentation faces limitations in complex scenarios when relying solely on RGB images, such as low-light conditions and occluded objects. To address these challenges, researchers have introduced multimodal inputs like depth and thermal images. Existing approaches primarily focus on two directions: cross-modal alignment and fusion strategies at various network levels [3, 6, 20, 8, 21, 22, 10, 11, 13, 23], and developing specialized feature extraction architectures for multimodal data [24, 5, 4, 25, 26]. However, these methods often suffer from increased model complexity due to

modality-specific networks and are constrained by the scarcity of large-scale multimodal datasets for pretraining.

## 2.2 Multimodal prompting

Multimodal prompting has emerged as an effective technique to enhance cross-modal reasoning in vision-language models. Visual prompts, complementing textual prompts, enable pixel-level instructions that help mitigate challenges like visual hallucinations and linguistic biases. Models like CLIP [27] and LLaVA [16] demonstrate how visual prompts combined with large language models can achieve strong performance in multimodal tasks. However, early applications in semantic segmentation, such as Dong *et al.* [14], often fuse multimodal features through simple operations like summation without adequately addressing inter-modal differences, which may interfere with RGB feature distributions and limit visual information capture. The complete literature review, including extensive analysis of prior research and additional references, can be found in the supplementary material (Section A.1).

## 3    Rethinking Multimodal Prompted Segmentation

**Problem setup.** Prompt tuning is a technique originally designed to improve the performance of a pretrained language model on a target task without modifying its internal architecture. It involves providing the model with task-specific context through properly constructed prompts. In the field of computer vision, similar ideas have emerged to adapt pretrained models to downstream tasks [28, 29]. A general workflow for this procedure can be formulated as follows:

$$e_i = P_i(h_{i-1}, e_{i-1}), \quad h_i = L_i(h_{i-1}, e_i), \quad y = \text{head}(h_N), \tag{1}$$

where $i \in \{1, 2, \ldots, N\}$ is the layer index. $P_i$ and $L_i$ denote the $i$-th prompting module and model layer, respectively. $h_i$ represents the output hidden states of layer $L_i$, and $h_0$ is the embedded original input. $e_i$ denotes the prompt for layer $L_i$, while $e_0$ is the initial prompt at the input stage, which can be either learnable or task-specific parameters. Under this framework, multimodal prompted segmentation can be achieved by using a pretrained single-modality model to extract hidden states from an RGB image, where the prompt $e_i$ is incorporated into the input space at each layer $L_i$. The initial prompt $e_0$ is obtained based on the auxiliary modality input, and the intermediate prompt $e_i$ is generated by the prompting module $P_i$, which takes both the previous hidden state $h_{i-1}$ and prompt $e_{i-1}$ as inputs. Finally, the result of the segmentation $y$ is obtained by processing the hidden state $h_N$ using the prediction head.

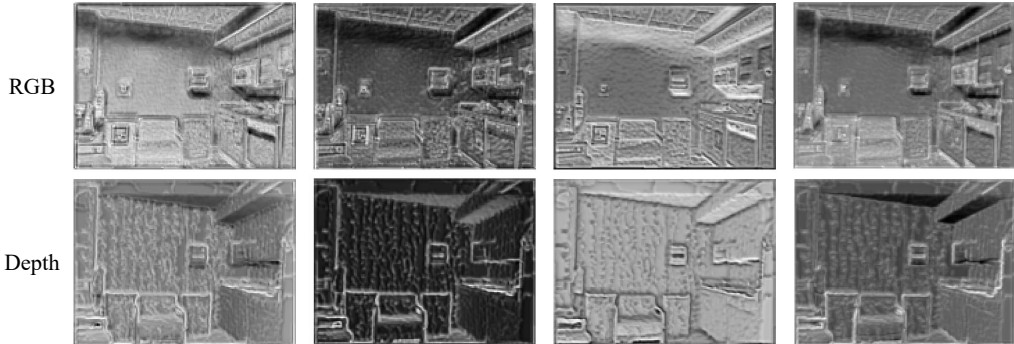

Figure 2: Feature maps extracted from the first stage of a pretrained RGB ResNet50 for RGB and depth inputs, showing that RGB pretrained model can extract meaningful features from depth images.

**Rethinking prompt initialization.** In the multimodal prompted segmentation framework described above, the way in which auxiliary modality information is introduced into a pretrained model significantly impacts segmentation performance, specifically the initial prompt $e_0$, as it serves as the entry point for incorporating the auxiliary modality into the model. We reconsider the typical approach to prompt initialization, which often relies on training a new feature extractor specific to the auxiliary modality or using randomly initialized prompts. This random initialization may negatively

affect the final performance of the prompted model. In contrast, we hypothesize that **early layers of a pretrained RGB model could provide a more effective initialization**.

To validate this, we conducted experiments to explore the viability of using pretrained RGB models for initializing the prompt $e_0$. Specifically, we analyze the feature maps obtained from the first stage of a pretrained ResNet50 model, originally trained on RGB data, using both RGB and depth modality inputs. For this analysis, we use the first sample from the test set of the NYU Depth V2 dataset, which consists of an RGB-depth image pair.

We begin by comparing the feature maps obtained from the model when fed with both RGB and depth inputs. Figure 2 presents a subset of the extracted feature maps for both modalities. These maps show that **both RGB and depth inputs exhibit distinct but rich structural patterns**, with each modality focusing on different aspects of the scene. For instance, the feature maps from the RGB input capture fine edge and texture details, while the depth modality emphasizes depth-related information, such as surface contours and object shapes. Despite the differences in data characteristics, the pretrained RGB model is still capable of extracting meaningful features from the depth modality.

## 4 Method

### 4.1 MixPrompt framework

We introduce the MixPrompt framework, a method designed to integrate auxiliary modality information into a pretrained RGB model through efficient and lightweight prompt tuning. As illustrated in Figure 3, our framework utilizes a pretrained RGB model as the backbone for processing RGB images. The auxiliary modality is incorporated via a lightweight prompting module that generates the initial prompt $e_0$ based on features extracted from the auxiliary input. At each layer $i$, the backbone layer $L_i$ and the prompting module $P_i$ iteratively fuse and refine features from the hidden state $h_{i-1}$ and the prompt $e_{i-1}$. The final output is then passed through a segmentation head—initialized with a pretrained RGB segmentation model—to generate the segmentation mask. During training, the backbone remains frozen, as it accounts for the majority of the model parameters. Therefore, only a small subset of parameters is trainable in the MixPrompt framework, ensuring efficient optimization.

The design of prompt initialization stems from our rethinking of how the initial prompt $e_0$ should be generated. Instead of training an additional modality-specific feature extractor, we propose leveraging the early layers of the pretrained RGB model. These layers are effective for initializing $e_0$, as they capture general, low-level features that can transfer across different modalities. In the following sections, we will detail the prompting process after obtaining the initial prompt $e_0$.

### 4.2 Multi-subspace prompting

We delve into the design of the prompting module in this subsection. In our MixPrompt framework, the prompting module serves as the core, fusing useful information from the auxiliary modality into the backbone. This integration ensures that the final prediction takes the auxiliary modality into account, leading to superior results. However, the features of the main modality in the backbone and those of the auxiliary modality in the prompting branch exist in different feature spaces. Consequently, the prompting module must first align these mismatched feature spaces. Once aligned, the features are fused and fed back into the backbone. This alignment and fusion process relies on two key design principles of our prompting module. The first principle is **parameter-efficient feature alignment**, which ensures that the prompting branch remains lightweight. The second principle is **information-exploitation efficiency** which ensures that we fully utilize the auxiliary information available. The designed prompting module, based on these two principles, is presented at the bottom right of Figure 3.

To achieve parameter-efficient feature alignment, it is essential that the alignment module remains uncomplicated. Therefore, we employ a straightforward linear projection to align the features between the RGB and auxiliary modalities within a low-rank subspace. This approach not only simplifies the alignment process but also enhances computational efficiency by reducing the complexity of the module. Furthermore, the fusion of these aligned features is implemented by simply adding them together, which further promotes efficient computation. This process is formulated as follows:

$$e_i = P_i(h_{i-1}, e_{i-1}) = W_{\text{up}}(W_{\text{rgb}}h_{i-1} + W_{\text{x}}e_{i-1}), \tag{2}$$

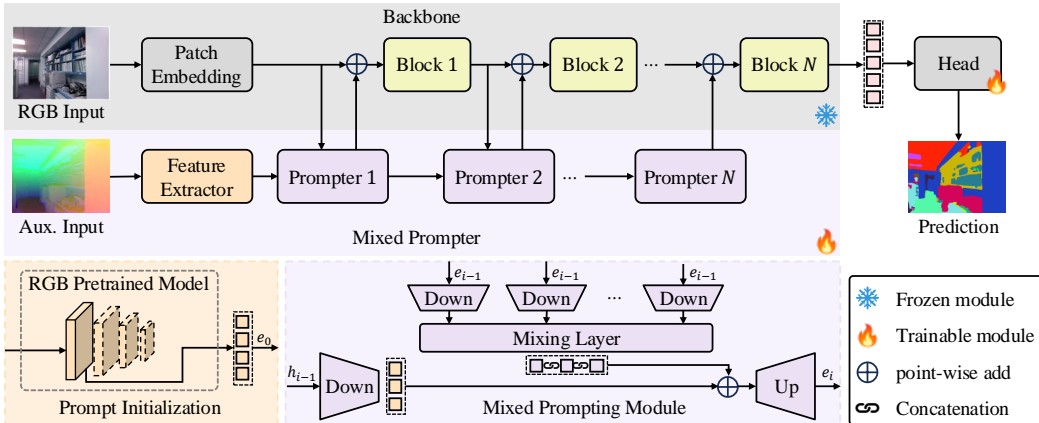

Figure 3: Overall framework of the proposed MixPrompt for multimodal semantic segmentation. The RGB input is processed by an RGB-pretrained segmentation model, while the auxiliary input is processed by a lightweight prompting module. In the prompting module, the initial prompt is derived from the early blocks of the pretrained RGB model (Section 3). At each stage, auxiliary features are projected into multiple subspaces and subsequently mixed to align with the RGB features by mixed prompting modules (Section 4.2). The fused features are used as prompt and is fed back into the main branch. Finally, the segmentation mask is generated by a prediction head based on the output features of the main branch. During training, only the lightweight prompting module and the prediction head are trainable, ensuring the parameter efficiency of the proposed framework.

where $W_{\text{rgb}}, W_{\text{x}} \in \mathbb{R}^{r \times d}$ are down projections and $W_{\text{up}} \in \mathbb{R}^{d \times r}$ is up projection. Here, $d$ is the original feature space, and $r \ll d$ is the dimension of the low-rank alignment subspace. By adopting a small value for $r$, the prompting module can be made parameter-efficient.

To enhance information-exploitation efficiency, we propose introducing multiple pairs of down and up projections. Each pair aligns the RGB and auxiliary modalities within a distinct subspace, thereby boosting the utilization of useful information. However, this approach increases the number of parameters significantly. Drawing inspiration from the LoRA finetuning method for LLMs [30], we observe that employing an extremely low rank can still yield promising performance. This is because the pretrained LLM operates within a low intrinsic dimension. We make a similar assumption: our pretrained RGB segmentation model also resides in a low intrinsic dimension. Consequently, we reduce the rank of each subspace. This reduction ensures that the overall number of parameters remains unchanged, thereby maintaining parameter efficiency. Taking the RGB features as an example, the projection into multiple subspaces can be represented as:

$$[W_{\text{rgb},1}h_{i-1}, W_{\text{rgb},2}h_{i-1}, ..., W_{\text{rgb},n}h_{i-1}], \tag{3}$$

where $W_{\text{rgb},1}, W_{\text{rgb},2}, ..., W_{\text{rgb},n} \in \mathbb{R}^{\frac{r}{n} \times d}$. $n$ indicates the number of adopted subspaces, where the division of the original rank $r$ by $n$ ensures that the introduction of multiple subspaces does not bring additional parameters. To keep concise, we denote $W_{\text{rgb},n}h_{i-1}$ as $h_{i-1}^n$. Then, to further improve the representation ability, we introduce a mixing matrix $M \in \mathbb{R}^{n \times n}$ to exchange information between each subspace. The mixed feature $\hat{h}_{i-1}$ becomes:

$$\hat{h}_{i-1} = \left[h_{i-1}^1, h_{i-1}^2, ..., h_{i-1}^n\right] M = \left[\sum_{j=1}^n M_{j,1}h_{i-1}^j, ..., \sum_{j=1}^n M_{j,n}h_{i-1}^j\right], \tag{4}$$

where $M_{i,j}$ indicates value of the according element in the mixing matrix $M$. Similarly, we obtain the low-rank representation of the prompt $e_{i-1}$ as $\hat{e}_{i-1}$. The information from the RGB modality is then injected into the low-rank prompt by addition and back-projected into the original feature space via the linear projection $W_{\text{up}} \in \mathbb{R}^{d \times r}$ to yield the new prompt $e_i$. Once the prompt $e_i$ is obtained, the backbone layer $L_i$ directly fuses the RGB feature with the prompt via addition. The whole process can be formulated as:

$$h_i = L_i(h_{i-1} + W_{\text{up}}(\hat{h}_{i-1} + \hat{e}_{i-1})). \tag{5}$$

The mixed prompting module performs information mixing only between different subspaces of the RGB modality, based on the rationale that mixing information across subspaces on both modalities

Table 1: Comparison of various multi-modal fusion methods on RGB-D segmentation datasets. Results are obtained through multiscale testing. '-' indicates that the corresponding results are not provided in the original paper. '*' denotes the result from the the single-modal version of the method using only RGB input.

| Model | Backbone | Params | NYU Depth V2 | | | SUN-RGBD | | |
|---|---|---|---|---|---|---|---|---|
| | | | Input size | FLOPs | mIoU | Input size | FLOPs | mIoU |
| SegFormer[*] [34] | MiT-B4 | 62.4M | 480×640 | 74.7G | 52.3 | 530×730 | 96.3G | 49.1 |
| ACNet [3] | ResNet50 | 116.6M | 480×640 | 126.7G | 48.3 | 530×730 | 163.9G | 48.1 |
| SGNet [4] | ResNet101 | 64.7M | 480×640 | 108.5G | 51.1 | 530×730 | 151.5G | 48.6 |
| SA-Gate [5] | ResNet101 | 110.9M | 480×640 | 193.7G | 52.4 | 530×730 | 250.1G | 49.4 |
| GEN [6] | ResNet101 | 118.2M | 480×640 | 118.2G | 51.7 | 530×730 | 790.3G | 50.2 |
| ShapeConv [24] | ResNext-101 | 86.8M | 480×640 | 124.6G | 51.3 | 530×730 | 161.8G | 48.6 |
| ESANet [22] | ResNet34 | 31.2M | 480×640 | 31.2G | 50.3 | 480×640 | 34.9G | 48.2 |
| TokenFusion [21] | MiT-B3 | 45.9M | 480×640 | 94.4G | 54.2 | - | - | - |
| TransD-Fusion [20] | Swin-B | 84.0M | 480×640 | - | 55.5 | 530×730 | - | 51.9 |
| Omnivore [25] | Swin-B | 95.7M | 480×640 | 95.7G | 54.0 | - | - | - |
| CMX [7] | MiT-B2 | 66.6M | 480×640 | 67.6G | 54.4 | 530×730 | 86.3G | 49.7 |
| CMX [7] | MiT-B4 | 139.9M | 480×640 | 134.3G | 56.3 | 530×730 | 173.8G | 52.1 |
| CMX [7] | MiT-B5 | 181.1M | 480×640 | 167.8G | 56.9 | 530×730 | 217.6G | 52.4 |
| CMNext [8] | MiT-B4 | 119.6M | 480×640 | 131.9G | 56.9 | - | - | - |
| DFormer [26] | DFormer-L | 39.0M | 480×640 | 65.7G | 57.2 | 530×730 | 83.3G | 52.5 |
| DPLNet [14] | MiT-B5 | 88.58M | 480×640 | 105.0G | 59.3 | 530×730 | 132.9G | 52.8 |
| Ours | MiT-B5 | 87.2M | 480×640 | 109.0G | **61.2** | 530×730 | 137.9G | **53.5** |

could hinder alignment and lead to suboptimal performance. The overall workflow of is presented in the supplementary material (Algorithm 1).

**Analysis.** The proposed prompting method adopts multiple subspaces for feature fusion, which acts somewhat like the adaptation of a mixture of experts in LLM design [31, 32], where matrix $M$ is the router to assign different importance to each subspace by mixing them with different weights. When $n = 1$ or when all elements in $M$ are equal to $1/n$, the prompting module downgrades to the base architecture in Equation 2. More quantitative analysis in the supplementary material (Section C.5) further validates this design, showing that moderate numbers of subspaces yield optimal performance by enhancing inter-subspace diversity. Another impact of the mixed prompting module comes from reparameterization [33]. While the introduction of multiple subspaces does not involve any non-linear operations, it allows the model to leverage a multi-branch architecture. This modification of the gradient flow helps the model achieve better performance by facilitating more effective information propagation. By aligning features in a low-rank subspace, we adhere to the design principle of parameter-efficient feature alignment. Additionally, the introduction of multi-subspace prompting and the information mixing scheme ensures the principle of information-exploitation efficiency, enabling the final model to achieve effective and efficient multimodal fusion.

## 5 Experiments

### 5.1 Experimental Setup

To validate the effectiveness of MixPrompt, we conduct experiments on multiple datasets with RGB and auxiliary modalities, including RGB-Depth, RGB-Thermal, RGB-Event, and RGB-Lidar. NYU Depth V2 [15] dataset contains 1,449 RGB-D samples across 40 categories. SUN-RGBD [19] includes 10,335 RGB-D images with 38 classes. MFNet [10] provides 1,569 RGB-Thermal pairs from 9 classes. DELIVER [8] comprises 3,983 training and 2,005 testing samples with RGB, Depth, Event, and Lidar modalities across 25 categories.

We use the mean Intersection over Union (mIoU) metric for evaluation, with multiscale testing on NYU Depth V2 and SUN-RGBD, and single-scale testing on other datasets, aligning with the evaluation practices of prior works. For the backbone network, we utilize the Mix Transformer encoder (MiT) [34] pretrained on the ADE20K dataset [2]. The auxiliary modalities are projected into the feature space using modules placed before the second stage of a ResNet50 [35] network, which initializes the prompt information. This prompt is then fused across multiple scales using four mixed

Table 2: Comparison of RGB-T segmentation performance on MFNet. Results marked with an underline show the second-best performance in each category, while those in bold indicate the highest score for that class. UL: Unlabeled, PS: Person, CT: Car top, GD: Guardrail, CC: Color cone.

| Method | Backbone | Params | FLOPs | UL | Car | PS | Bike | Curve | CT | GD | CC | Bump | mIoU |
|---|---|---|---|---|---|---|---|---|---|---|---|---|---|
| MFNet [10] | - | - | - | 96.9 | 65.9 | 58.9 | 42.9 | 29.9 | 9.9 | 0.0 | 25.2 | 27.7 | 39.7 |
| RTFNet [11] | ResNet152 | 245.7M | 185.2G | _98.5_ | 87.4 | 70.3 | 62.7 | 45.3 | 29.8 | 0.0 | 29.1 | 55.7 | 53.2 |
| PSTNet [12] | ResNet18 | 105.8M | 123.4G | 97.0 | 76.8 | 52.6 | 55.3 | 29.6 | 25.1 | **15.1** | 39.4 | 45.0 | 48.4 |
| FuseSeg [13] | DenseNet161 | 141.5M | 193.4G | 97.6 | 87.9 | 71.7 | 64.6 | 44.8 | 22.7 | 6.4 | 46.9 | 47.9 | 54.5 |
| U2Fusion [36] | VGG16 | - | - | 97.7 | 82.8 | 64.8 | 61.0 | 32.3 | 20.9 | - | 45.2 | 50.2 | 50.8 |
| AFNet [37] | ResNet50 | - | - | 98.0 | 86.0 | 67.4 | 62.0 | 43.0 | 28.9 | 4.6 | 44.9 | 56.6 | 54.6 |
| ABMDRNet [38] | ResNet50 | 64.6M | 194.3G | **98.6** | 84.8 | 69.6 | 60.3 | 45.1 | 33.1 | 5.1 | 47.4 | 50.0 | 54.8 |
| FEANet [39] | ResNet152 | 337.1M | 255.2G | 98.3 | 87.8 | 71.1 | 61.1 | 46.5 | 22.1 | 6.6 | _55.3_ | 48.9 | 55.3 |
| GMNet [40] | ResNet50 | 149.8M | 153.0G | 97.5 | 86.5 | 73.1 | 61.7 | 44.0 | _42.3_ | _14.5_ | 48.7 | 47.4 | 57.3 |
| TarDAL [41] | - | 297M | - | 97.6 | 80.7 | 67.1 | 60.1 | 34.9 | 10.5 | - | 38.7 | 45.5 | 48.6 |
| EAEFNet [42] | ResNet152 | 200.4M | 147.3G | - | 87.6 | 72.6 | 63.8 | 48.6 | 35.0 | 14.2 | 52.4 | 58.3 | 58.9 |
| CACFNet [43] | ConvNeXt-B | 198.6M | 101.4G | - | 89.2 | 69.5 | 63.3 | 46.6 | 32.4 | 7.9 | 54.9 | 58.3 | 57.8 |
| PAIF [44] | - | 260M | - | 88.1 | 72.4 | 48.1 | 60.8 | - | - | - | **56.0** | 57.2 | 56.5 |
| CENet [23] | ResNet50 | - | - | 98.1 | 87.8 | 71.4 | 63.2 | 47.5 | 31.1 | - | 48.9 | 50.3 | 56.1 |
| SegMiF [45] | MiT-B3 | - | - | 98.1 | 87.8 | 71.4 | 63.2 | 47.5 | 31.1 | - | 48.9 | 50.3 | 56.1 |
| CMX [7] | MiT-B2 | 66.6M | 67.6G | 98.3 | 89.4 | 74.8 | 64.7 | 47.3 | 30.1 | 8.1 | 52.4 | _59.4_ | 58.2 |
| CMX [7] | MiT-B4 | 139.9M | 134.3G | 98.3 | 90.1 | _75.2_ | 64.5 | _50.2_ | 35.3 | 8.5 | 54.2 | **60.6** | 59.7 |
| CMNeXt [8] | MiT-B4 | 119.6M | 131.9G | 98.4 | **91.5** | **75.3** | **67.6** | **50.5** | 40.1 | 9.3 | 53.4 | 52.8 | _59.9_ |
| DPLNet [14] | MiT-B5 | 88.58M | 105.0G | - | - | - | - | - | - | - | - | - | 59.3 |
| Ours | MiT-B5 | 87.2M | 109.0G | 98.3 | _90.2_ | 74.5 | _65.2_ | 50.1 | **48.3** | 10.5 | 51.7 | 52.0 | **60.1** |

prompting modules, effectively integrating the auxiliary information into the RGB backbone at various levels. Additional optimization details are provided in the supplementary material (Section C.1).

## 5.2 Segmentation results

### 5.2.1 RGB-D segmentation

We first analyze the performance of the proposed MixPrompt framework for RGB-D (depth) segmentation, comparing it with several state-of-the-art multimodal fusion methods. The results are summarized in Table 1, which reports the performance on the NYU Depth V2 and SUN-RGBD datasets.

On the NYU Depth V2 dataset, MixPrompt achieves the highest mIoU score of 61.2 with 87.2M parameters and 109.0G FLOPs computational cost, demonstrating the efficiency of our approach. In comparison, the second-best method, DPLNet, achieves an mIoU of 59.3 with slightly more parameters and lower FLOPs. Methods like ACNet and SGNet, with mIoUs of 48.3 and 51.1, respectively, perform significantly worse, highlighting the advantages of our MixPrompt.

On the SUN-RGBD dataset, MixPrompt again achieves the highest mIoU of 53.5, surpassing other methods by a notable margin. DPLNet ranks second with an mIoU of 52.8, while CMX-MiT-B5 follows closely with 52.4. Other methods, such as ACNet and SGNet, continue to show lower performance, with mIoUs of 48.1 and 48.6, respectively. The results on SUN-RGBD further reinforce the strength of MixPrompt. A key advantage of MixPrompt is its computational efficiency. With 87.2M parameters and 109.0G FLOPs on the NYU Depth V2 dataset, MixPrompt achieves competitive accuracy while being more lightweight compared to other high-performance models, such as CMX (MiT-B5). The efficient design of the MixPrompt framework, using a lightweight prompting module integrated with a pretrained RGB backbone, avoids the need for parallel networks or complex fusion architectures that typically increase both model size and computational cost. Another notable method is DFormer [26], which has significantly fewer parameters compared to other recent approaches. While its design of using a shared branch for both RGB and auxiliary modalities contributes to a parameter-efficient architecture, this shared parameterization also limits flexibility of the model. As a result, DFormer significantly underperforms in comparison to our framework.

To ensure the fairness of our experimental setup, we additionally examined the impact of different backbone pretraining settings. Specifically, we re-trained several representative multi-modal methods (e.g., CMX and CMNeXt) using the same ADE20K-pretrained weights as ours to provide an equitable

comparison. The results consistently show that while these models benefit slightly from stronger initialization, our method still achieves a clear advantage (around 2.9–3.1 mIoU higher on NYUDepth) with significantly fewer trainable parameters. Detailed experimental results and analysis are provided in the supplementary material (Section C.3).

### 5.2.2 RGB-T segmentation

We then analyze the performance of MixPrompt on the RGB-T (Thermal) segmentation dataset MFNet. The results, shown in Table 2, report both class-wise and overall mIoU.

On the overall mIoU metric, MixPrompt achieves the highest score of 60.1, outperforming all other models in the comparison. The second-best method, CMNeXt, achieves an mIoU of 59.9. These top performers are followed by other notable methods such as CMX (MiT-B4), which scored 59.7. In contrast, models like MFNet and RTFNet achieved significantly lower mIoUs of 39.7 and 53.2, respectively, showcasing the advantages of the MixPrompt framework. Focusing on the class-wise performance, our model performs exceptionally well on classes such as "Car" and "Bike", with scores of 90.2 and 74.5, respectively. These scores are higher than those achieved by the best-performing models, such as CMNeXt, which scored 91.5 and 75.3 for the same classes.

Notably, our model excels in the Car Top (CT) class, achieving an outstanding performance of 48.3, which surpasses all other models in this category.

Despite the competitive performance in many classes, our model continues to demonstrate an overall advantage, achieving a balance between class-wise accuracy and computational efficiency. With 87.2M parameters and 109.0G FLOPs, MixPrompt outperforms others with comparable or even fewer parameters, while maintaining strong class performance.

### 5.2.3 Other auxiliary modalities

The results in Table 3 compare different methods on the RGB-E (Event) and RGB-L (Lidar) segmentation tasks from the DELIVER dataset. For the RGB-E task, our model achieves a notable mIoU score of 58.0, surpassing all other RGB-E models, including the second-best CMNeXt and CMX, demonstrating the efficiency and effectiveness of our approach in handling multi-modal RGB-E data.

In the RGB-L task, our model further establishes its superiority by achieving the highest mIoU of 59.1. It outperforms the next best method by a considerable margin. Despite having only 29.9M parameters, our model delivers top-tier performance, demonstrating both efficiency and strong segmentation capabilities in the RGB-L setting.

Overall, our model consistently performs at the top across both RGB-E and RGB-L modalities, showing a robust ability to integrate and process multimodal data for segmentation tasks. The relatively low number of parameters required further emphasizes the efficiency of our approach compared to other high-performing models.

Table 3: Comparison of RGB-Event and RGB-Lidar segmentation performance on the DELIVER dataset. The highest mIoU for each condition is highlighted in bold, while the second-best score is underlined.

| Method | Modal | Backbone | Params | mIoU |
|---|---|---|---|---|
| HRFuser [46] | RGB | HRFormer-T | 29.9M | 48.0 |
| CMNeXt [8] | RGB | MiT-B2 | 25.8M | 57.2 |
| HRFuser [46] | RGB-E | HRFormer-T | 30.5M | 42.2 |
| TokenFus. [21] | RGB-E | MiT-B2 | 26.0M | 45.6 |
| CMX [7] | RGB-E | MiT-B2 | 66.6M | 56.5 |
| CMNeXt [8] | RGB-E | MiT-B2 | 58.7M | 57.5 |
| Ours | RGB-E | MiT-B2 | 29.9M | **58.0** |
| HRFuser [46] | RGB-L | HRFormer-T | 30.5M | 43.1 |
| TokenFus. [21] | RGB-L | MiT-B2 | 26.0M | 53.0 |
| CMX [7] | RGB-L | MiT-B2 | 66.6M | 56.4 |
| CMNeXt [8] | RGB-L | MiT-B2 | 58.7M | 58.0 |
| Ours | RGB-L | MiT-B2 | 29.9M | **59.1** |

### 5.3 Ablation study

To comprehensively evaluate our approach, we conduct an ablation study on the NYU Depth V2 dataset using MiT-B5 as the default model. The ablations cover three main aspects: the effectiveness of each module, the detailed design choices within each module, and the impact of key hyperparameters.

### 5.3.1 Effectiveness of each module

In order to evaluate the contributions of the key components in our proposed method, we conduct an ablation study focusing on two essential modules: the pretrained prompt extractor and the mixed prompting module. The results are summarized in Table 4.

When both modules are enabled, the best performance, with an mIoU of 60.1%, is achieved. However, when the pretrained prompt extractor is disabled, where we employ a simple patch embedding layer to integrate the auxiliary modality, the model achieves an mIoU of 58.8%. Similarly, if the mixed prompting module is removed, modality fusion is achieved through direct addition of the features without any alignment, and the mIoU score decreases slightly to 59.5%. From these experiments, the combination of the pretrained prompt extractor and the mixed prompting module provides the best results, highlighting that both modules in our proposed framework play an important role in enhancing performance.

Table 4: Ablation results for the pretrained prompt extractor and the mixed prompting module.

| Prompt extractor | Mixed Prompting | Trainable params | FLOPs | mIoU (%) |
|:---:|:---:|:---:|:---:|:---:|
| ✓ | ✓ | 5.7M | 109.0G | **60.1** |
| ✗ | ✓ | 5.5M | 104.7G | 58.8 |
| ✓ | ✗ | 5.4M | 108.7G | 59.5 |

### 5.3.2 Prompt extractor

We further investigate the design of the pretrained prompt extractor, beginning with an analysis of whether initializing the extractor with pretrained weights affects performance. As shown in Table 5, using a randomly initialized ResNet50 prompt extractor results in an mIoU of 59.5%, whereas initializing it with pretrained weights improves performance to 60.1%. This highlights the effectiveness of leveraging pretrained RGB models for extracting initial prompts. The pretrained backbone enables the model to extract more meaningful representations from the auxiliary modality, ultimately leading to better segmentation results.

Table 5: Impact of initializing the ResNet50 prompt extractor with pretrained weights.

| Initialization type | mIoU (%) |
|:---|:---:|
| Random | 59.5 |
| Pretrained | **60.1** |

Next, we assess the impact of different prompt extractor architectures, with the results presented in Table 6. Given that the backbone follows the MiT architecture, we first evaluate various MiT variants as the prompt extractor. The results from MiT-B1 show that deeper extractors lead to lower mIoU, while progressively reducing the depth improves performance, with the best result achieved using only the first stage. Notably, ResNet50 outperforms MiT-B1, highlighting the effectiveness of convolutional models in extracting initial prompts. Additionally, increasing the size of MiT extractors offers no further improvements, reinforcing the advantage of convolutional extractors.

These findings suggest that a pretrained convolutional prompt extractor focusing on early-stage features is more beneficial for prompted segmentation.

Table 6: Impact of different prompt extractor architectures and the number of layers used. A shallower convolutional extractor focusing on early-stage features achieves the best results.

| Prompt extractor | Layers | mIoU (%) |
|:---:|:---:|:---:|
| mit-b1 | stage {1,2,3,4} | 59.2 |
| mit-b1 | stage {1,2,3} | 59.5 |
| mit-b1 | stage {1,2} | 59.6 |
| mit-b1 | stage {1} | 59.7 |
| mit-b2 | stage {1} | 59.6 |
| mit-b4 | stage {1} | 59.2 |
| mit-b5 | stage {1} | 59.3 |
| ResNet50 | stage {1,2,3,4} | 58.9 |
| ResNet50 | stage {1,2,3} | 59.5 |
| ResNet50 | stage {1,2} | 59.9 |
| ResNet50 | stage {1} | **60.1** |

The first stage of a pretrained ResNet50 provides the best performance. Therefore, we adopt this configuration as our final design.

### 5.3.3 Prompt mixing

We further investigate the impact of different prompt mixing configurations, particularly the effect of multi-subspace prompt mixing when applied to either the RGB or auxiliary modality. The results are summarized in Table 7.

From the results, the model achieves an mIoU of 59.3% without multi-subspace mixing. When multi-subspace mixing is introduced to the RGB modality, performance improves significantly to 60.1%, demonstrating the benefit of refining RGB prompts through multi-subspace alignment. Conversely, applying multi-subspace mixing only to the auxiliary modality yields a smaller improvement, reaching 59.7%. Interestingly, enabling multi-subspace mixing for both modalities does not lead to further gains and instead results in a slightly lower mIoU of 59.8% compared to the RGB-only setting.

Table 7: Ablation study on multi-subspace prompt mixing. The best result is achieved when multi-subspace mixing is applied only to RGB prompts.

| RGB mixing | Auxiliary mixing | mIoU (%) |
|:---:|:---:|:---:|
| ✗ | ✗ | 59.3 |
| ✓ | ✗ | **60.1** |
| ✗ | ✓ | 59.7 |
| ✓ | ✓ | 59.8 |

These results suggest that enhancing the RGB modality with multi-subspace mixing plays a more crucial role in improving segmentation performance. In contrast, applying the same strategy to the auxiliary modality provides more limited benefits, and simultaneous application to both may introduce redundant or conflicting information. Thus, our final design prioritizes multi-subspace mixing for RGB prompts to achieve optimal results.

### 5.3.4 Hyperparameters

We further perform an ablation study on key hyperparameters. First, we analyze the effect of the rank downscale ratio $\frac{d}{r}$, which controls the intermediate feature dimension and effectively determines the rank of the low-rank subspace. A larger ratio corresponds to a smaller rank and fewer trainable parameters. As shown in Table 8, increasing the downscale ratio from 1 to 4 improves the mIoU from 59.4% to 60.1%, indicating that a moderate reduction in rank enhances feature efficiency. However, when the ratio is further increased to 8, performance drops slightly to 59.6%, suggesting that an excessively small rank may lead to information loss and hinder performance. The optimal trade-off is achieved at a ratio of 4.

Next, we investigate the impact of the number of subspaces $n$ used for prompt mixing. Unlike the rank downscale ratio, this parameter does not affect parameter efficiency of the model. As reported in Table 9, increasing the number of subspaces from 1 to 4 progressively enhances performance, reaching a peak mIoU of 60.1%. However, further increasing the number of subspaces to 8 results in a performance decline, suggesting that overly complex prompt mixing may introduce unnecessary redundancy.

Furthermore, we validated our hyperparameter configuration on multiple multimodal datasets, including RGB-Thermal, RGB-LiDAR, and RGB-Event. The results indicate that the selected values strike an effective balance among model capacity, generalization, and computational efficiency. Due to space limitations, more detailed results are provided in the supplementary material (Section C.7).

Table 8: Ablation study on the rank downscale ratio.

| Rank down scale ratio $\frac{d}{r}$ | 1 | 2 | 4 | 8 |
|:---:|:---:|:---:|:---:|:---:|
| mIoU (%) | 59.4 | 59.8 | **60.1** | 59.6 |

Table 9: Ablation study on the number of subspaces for prompt mixing.

| Num. of Subspace $n$ | 1 | 2 | 4 | 8 |
|:---:|:---:|:---:|:---:|:---:|
| mIoU (%) | 59.3 | 59.8 | **60.1** | 59.3 |

## 6 Conclusion

In this paper, we introduced MixPrompt, a novel framework for multimodal semantic segmentation that efficiently integrates auxiliary modalities into pretrained RGB models through prompt tuning. By leveraging a lightweight prompting module and multi-subspace alignment, MixPrompt successfully enhances model performance while maintaining parameter efficiency. Our method addresses the challenges of increased model complexity and data scarcity commonly associated with multimodal segmentation tasks. Through comprehensive experiments on multiple datasets, including NYU Depth V2, SUN-RGBD, MFNet, and DELIVER, we demonstrated that MixPrompt outperforms existing dual-backbone approaches with fewer parameters, establishing it as a highly effective and scalable solution.

## Acknowledgment

This work was supported by the Joint Funds of the National Natural Science Foundation of China (NSFC) (No. U2336211), the Major Research Plan of the NSFC (No. 92467206), and the NSFC Grant (No. 62576364).

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

# A Related works

## A.1 Semantic segmentation.

The task of semantic segmentation not only requires identifying objects within an image but also demands precise outlining of their boundaries, rendering them considerably more challenging compared to image classification tasks. The development of traditional methods can be divided into two major stages. The first stage revolves around classic approaches based on handcrafted features, ranging from k-means clustering [47] to Markov Random Fields [48] and Conditional Random Fields [49]. The second stage marks the shift to early deep learning-based methods, such as Fully Convolutional Networks [50], which laid the foundation for modern segmentation tasks. Building on this foundation, techniques like multi-scale pyramids [51], attention mechanisms [52], and dilated convolutions [53] have been introduced, significantly improving segmentation accuracy by capturing richer contextual information and enhancing feature representation.

Although these methods have achieved commendable performance, they typically rely on RGB images for predictions. However, RGB images alone often lack sufficient information for precise semantic segmentation, especially in complex scenarios such as low-light conditions, blurred textures, or occluded target objects. To address these challenges, researchers are exploring the integration of other data modalities, such as depth maps [15, 19] and Thermal images [10], leveraging the complementary advantages of multimodal data to enhance the accuracy and robustness of semantic segmentation across diverse environments.

Existing methods primarily focus on two aspects. The first aspect emphasizes the alignment and fusion between different modalities [3, 6, 20, 8, 21, 22, 10, 11, 13, 23]. Researchers explored diverse strategies for alignment and fusion across multiple levels, encompassing the input, intermediate feature extraction layers, and output. For instance, Cao *et al.* [24] incorporates geometric information from auxiliary modalities into convolutional weights, establishing a link between the weights and the underlying spatial relationships of corresponding pixels to better capture the spatial structure of scenes. Similarly, Hu *et al.* [3] introduces an additional attention-based auxiliary module to fuse features from different modalities, further balancing feature distributions and enabling the network to focus more effectively on the most relevant regions of the image. Zhang *et al.* [7] proposes innovative cross-modal feature calibration and fusion modules, aligning and calibrating feature differences in spatial and channel dimensions across modalities at multiple scales of the model.The second aspect focuses on more effective feature extraction [24, 5, 4, 25, 26]. For example, Girdhar *et al.* [25] introduces a novel Transformer architecture that is jointly pre-trained on images, videos, and single-view 3D data, equipping the model with cross-modal semantic feature extraction capabilities and making it suitable for downstream tasks across different modalities. Meanwhile, Zhang *et al.* [26] constructs a new large-scale RGB-D image dataset for pre-training, enhancing the ability of the model to encode both RGB and depth images.

Nevertheless, the introduction of additional modality-specific feature extraction networks significantly increases model complexity, resulting in an overwhelming training overhead. Moreover, the scarcity of existing multimodal datasets makes it challenging to support large-scale pretraining tasks, which hinders the model ability to be quickly fine-tuned for various downstream tasks across different modalities.

## A.2 Multimodal Prompting.

Vision-Language Models (VLMs) are deep learning models that integrate both visual and textual information, with visual capabilities that enable understanding and reasoning across complex multimodal tasks. The introduction of visual prompts, complementing textual prompts, enables more granular, pixel-level instructions on multimodal inputs, helping to mitigate challenges in traditional multimodal language models, such as visual hallucinations [54] and linguistic biases [55]. For instance, CLIP [27] leverages contrastive learning to align shared semantic spaces between images and text. LLaVA [16], by combining visual prompts with large-scale pretrained language models, has achieved remarkable results in image-text reasoning and generation tasks. This approach not only taps into the powerful foundational capabilities of large-scale pretrained models but also allows visual prompts to adaptively establish connections between different modalities in various forms tailored to specific tasks. An initial attempt to apply prompt tuning to the multimodal semantic segmentation task was made by Dong *et al.* [14]. However, their approach utilizes the same summation operation

to fuse features from different modalities without considering the differences between the additional modality and the RGB modality, which may interfere with the feature distribution of the primary RGB modality and affect ability of the model to capture important visual information.

# B  Methodology Details

## B.1  Details of mixed prompting

The detailed algorithmic procedure for the mixed prompting module is presented in Algorithm 1.

---

**Algorithm 1** Mixed Prompting Module

---

**Require:** hidden state $h_{i-1}$, prompt $e_{i-1}$, low-rank matrices $W_{\text{rgb}} \in \mathbb{R}^{r \times d}$, $W_{\text{x}} \in \mathbb{R}^{r \times d}$, $W_{\text{up}} \in \mathbb{R}^{d \times r}$, number of subspaces $n$, mixing matrix $M \in \mathbb{R}^{n \times n}$, model layer $L_i$
**Ensure:** hidden state $h_i$, prompt $e_i$
1:  $\hat{e}_{i-1} \leftarrow W_{\text{x}} e_{i-1}$
2:  $\hat{h}_{i-1} \leftarrow W_{\text{rgb}} h_{i-1}$
3:  $\hat{h}_{i-1} \leftarrow \text{reshape}(\hat{h}_{i-1}, (-1, n))$
4:  $\hat{h}_{i-1} \leftarrow \hat{h}_{i-1} M$             ▷ *Mixing*
5:  $\hat{h}_{i-1} \leftarrow \text{flatten}(\hat{h}_{i-1})$
6:  $e_i \leftarrow W_{\text{up}}(\hat{e}_{i-1} + \hat{h}_{i-1})$        ▷ *new prompt*
7:  $h_i \leftarrow L_i(h_{i-1} + e_i)$        ▷ *new hidden state*

---

# C  Additional Experiment Details.

## C.1  Optimization and Schedule.

We provide detailed configurations for training on datasets including NYU Depth V2 [15], SUN-RGBD [19], MFNet [10], and DELIVER [8] datasets. The specific hyperparameters are summarized in Table 10.

For the *NYU Depth V2* dataset, we use SGD [56] with a weight decay of $5 \times 10^{-4}$ and an initial learning rate of 0.04. The model is trained for 500 epochs with a batch size of 8. For the *SUN-RGBD* dataset, we adopt the AdamW optimizer [57] with 100 epochs and an initial learning rate of 0.005. For the *MFNet* dataset, we train for 500 epochs with the AdamW optimizer, a learning rate of $6 \times 10^{-4}$, and a batch size of 4. For the *DELIVER* dataset, we train for 200 epochs with a batch size of 2, using a learning rate of $6 \times 10^{-5}$ and a weight decay factor of 0.01. All experiments are conducted on NVIDIA GeForce RTX 3090 GPUs. Data augmentation techniques, including random flipping, random cropping, and multiscale inference with scales {0.5, 0.75, 1.0, 1.25, 1.5, 1.75}, are applied during training for all datasets.

Table 10: Training configurations for different datasets. LR: Learning Rate, WD: Weight Decay, Mom: Momentum.

| Dataset | Input Size | Batch Size | Epochs | Optimizer | LR | WD | Mom |
|---------|-----------|-----------|--------|-----------|-----|-----|-----|
| NYUD-v2 | 480×640 | 8 | 500 | SGD | 4e-2 | 0.0005 | 0.9 |
| SUN-RGBD | 530×730 | 4 | 100 | AdamW | 5e-3 | 0.01 | (0.9, 0.999) |
| MFNet | 480×640 | 4 | 500 | AdamW | 6e-4 | 0.01 | (0.9, 0.999) |
| DELIVER | 1024×1024 | 2 | 200 | AdamW | 6e-5 | 0.01 | (0.9, 0.999) |

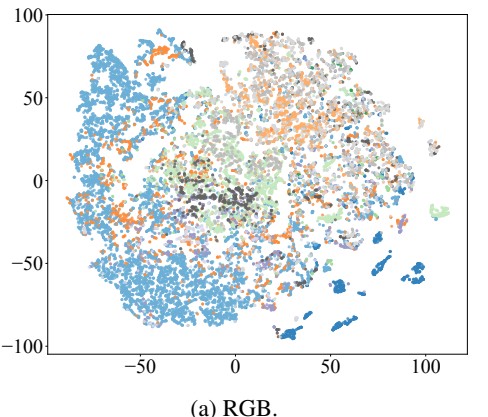 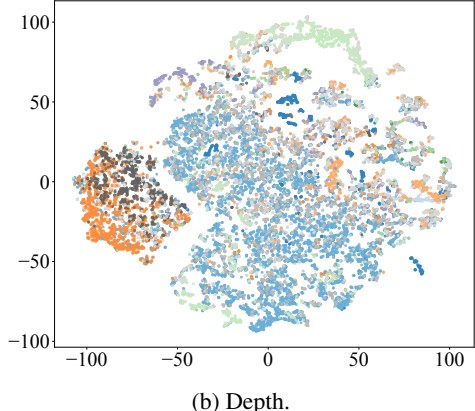

(a) RGB.    (b) Depth.

Figure 4: t-SNE visualization of feature embeddings for RGB and depth inputs, indicating that the feature from depth images results in similar clustering levels to those from RGB images.

## C.2 Analysis for Prompt Initialization

To further validate the feasibility of using pre-trained RGB models for prompt initialization, we conduct a t-SNE analysis to visualize the distribution of features extracted from both modalities.

In this analysis, each pixel is treated as a data point, and the corresponding ground truth serves as the label. The results, presented in Figure 4, demonstrate that **the feature representations for both RGB and depth inputs form clusters at a similar level in the lower-dimensional space**. For the RGB modality, the points are spread out with multiple clusters, reflecting the rich variety of information captured from the RGB image. Similarly, in the depth modality, the points also form well-defined clusters. This suggests that the features extracted from the depth modality, although distinct from RGB features in some aspects, are sufficiently informative, indicating that pretrained RGB models allows MixPrompt to efficiently integrate auxiliary modalities without requiring additional, modality-specific pretrained extractors.

## C.3 Analysis for Different Backbone Pretraining Settings

This note addresses the concerns raised regarding the fairness of the experimental setup. To ensure a equitable comparison, we have conducted additional experiments where we initialized several prior multi-modal approaches (e.g., CMX and CMNeXt) with ADE20K-pretrained weights, followed by training on the NYUDepth dataset. The results are summarized below.

Table 11: Comparison of different method with different backbone pretraining settings.

| Method | Backbone Pretraining | Trainable Params | mIoU (%) |
|---|---|---|---|
| Segformer | MiT-B5 (ADE20K) | 82.7M | 54.7 |
| CMX | MiT-B5 (ImageNet) | 181.1M | 56.9 |
| CMX | MiT-B5 (ADE20K) | 181.1M | 58.1 |
| CMNeXt | MiT-B4 (ImageNet) | 119.6M | 56.9 |
| CMNeXt | MiT-B4 (ADE20K) | 119.6M | 58.3 |
| Ours | MiT-B5 (ADE20K) | **5.7M** | **61.2** |

As shown in Table 11, we observe that CMX and CMNeXt do benefit from using the same strong baseline as ours, resulting in limited performance gains. However, the gap between their results and ours remains substantial (e.g., a 2.9–3.1 mIoU difference on the NYUDepth dataset). This comparison under the same pretrained setting demonstrates the effectiveness of our method over other multi-modal fusion approaches. Notably, our model achieves this with only 5.7M trainable parameters—significantly fewer than the alternatives.

Additionally, we conducted experiments where the single-modality RGB backbone (SegFormer) was also initialized with ADE20K-pretrained weights. The results further indicate that the performance

gains achieved by our method go well beyond what can be attributed to pretraining alone, highlighting the strength of our fusion strategy.

Overall, these experiments provide meaningful evidence for our fairness of the experimental setup.

## C.4 Experiments on the Effectiveness of the Encoding Strategy

To evaluate the contribution of reusing the early layers of a pretrained RGB backbone as an embedding strategy for auxiliary modalities, we conducted additional experiments by replacing our pretrained RGB backbone with the convolution-based encoder proposed in [58]. Experiments were performed on both RGB-D and RGB-T datasets, and the results are summarized in Table 12.

Table 12: Comparison of different encoding methods in the Mixed Prompting Module.

| Encoder Method | NYUDepth (RGB-D) | MFNet (RGB-T) |
|---|---|---|
| Convolution-Based Encoding [58] | 58.7 | 56.8 |
| Ours | **61.2** | **60.1** |

As shown in Table 12, the previous convolution-based depth encoder does not adapt well to the prompting framework. It likely provides suboptimal prompt initialization, leading to an approximate performance drop of 2.5% mIoU compared to our trainable backbone on RGB-D datasets. Moreover, results on the RGB-T dataset reveal that it fails to generalize effectively to arbitrary modality encoders, limiting its scalability.

## C.5 Quantitative Analysis of multi-subspace distributions.

To quantitatively characterize the diversity among the learned subspaces, we conducted additional analysis. Following the hierarchical encoder architecture of SegFormer, our backbone extracts four-stage multi-scale features with progressively lower spatial resolutions, denoted as d1-d4 (from highest to lowest resolution). For each stage, we compute the average pairwise cosine similarity between projected RGB subspace features, where values range from -1 (completely dissimilar) to 1 (identical). Lower (more negative) similarity values indicate greater diversity across subspaces. The "Avg.Sim (overall)" column summarizes the mean similarity across all four stages, providing an overall measure of inter-subspace diversity.

Table 13: Average pairwise cosine similarity (Sim) between RGB subspace features across different numbers of subspaces (n) and corresponding segmentation performance.

| n | Sim (d1) | Sim (d2) | Sim (d3) | Sim (d4) | Sim (overall) | mIoU (%) |
|---|---|---|---|---|---|---|
| 1 | N/A | N/A | N/A | N/A | N/A | 59.3 |
| 2 | -0.0338 | -0.0355 | -0.0508 | -0.0853 | -0.0514 | 59.8 |
| 4 | -0.1434 | -0.1715 | -0.2031 | -0.2506 | -0.1922 | 60.1 |
| 8 | -0.0135 | -0.0046 | -0.0175 | -0.0321 | -0.0169 | 59.6 |

As reported in Table 13, we evaluated how varying the number of subspaces (n) affects feature diversity using RGB-D inputs from the NYU Depth V2 test set. Our analysis reveals that within the same n, similarity values become increasingly negative from high-resolution (d1) to low-resolution (d4) features. This occurs because high-resolution features predominantly capture modality-agnostic local patterns (e.g., edges and textures) that remain relatively consistent across subspaces, whereas lower-resolution features encode more abstract, global, and cross-modal semantics. Consequently, our multi-subspace prompt mixing more effectively disentangles these higher-level representations, resulting in greater divergence at coarser scales.

Furthermore, increasing n from 1 to 4 substantially reduces overall similarity (from -0.0514 to -0.1922), reflecting enhanced structural diversity that correlates with optimal segmentation performance (mIoU of 60.1%). However, further increasing n to 8 causes similarity magnitude to decrease (from -0.1922 to -0.0169), indicating over-fragmentation that diminishes subspace diversity and aligns with the observed performance degradation.

These quantitative results validate the core motivation behind our multi-subspace design: by introducing structural diversity and encouraging disentangled prompt composition, it effectively improves segmentation outcomes. While comprehensive visualizations of subspace distributions would provide additional intuitive insights, we believe this detailed statistical characterization offers compelling evidence for the effectiveness of our approach.

### C.6 Analysis for Merging Strategy in the Mixed Prompting Module

We adopt a simple summation of the two prompt embeddings ($h_{i-1}$ and $e_{i-1}$), primarily due to its parameter-efficiency and empirical effectiveness in our preliminary experiments. While our current approach is effective, we fully recognize the potential of alternative fusion strategies—such as concatenation followed by a projection layer, learnable weighted summation via gating, or attention-based fusion—to enable more expressive cross-modal interactions, as the reviewer rightly pointed out. However, these methods typically introduce additional parameters and computational overhead, potentially undermining the objective of preserving a compact and efficient prompting module, while also increasing the complexity of training.

To futher investigate the effectiveness of these fusion methods, we conducted ablation studies on different prompt fusion strategies using the NYUDepth dataset. The quantitative results are shown in the table below, and the implementation details of each variant are described as Table 14.

**Concatenation.** Concatenate $h_{i-1}$ and $e_{i-1}$ along the module dimension, and apply a linear transformation to fuse the combined representation back into the original embedding space.

**Learnable Weighted Summation.** Introduce a learnable gating vector $g \in \mathbb{R}^C$ and compute the fused prompt as $\sigma(g) \cdot h_{i-1} + (1 - \sigma(g)) \cdot e_{i-1}$, where $\sigma$ denotes the sigmoid activation function. The gate $g$ is initialized as a trainable parameter and applied channel-wise to adaptively modulate the contribution of each modality.

**Attention Fusion.** The fused prompt is obtained by computing cross-modal attention between $e_{i-1}$ and $h_{i-1}$ : $\text{Attn}(Q, K, V) = \text{softmax}\left(\frac{QK^\top}{\sqrt{d}}\right) V$.

Overall, our results on NYUDepth indicate that simple summation achieves the best performance among the evaluated strategies, highlighting its effectiveness as a lightweight and robust fusion mechanism. Although concatenation and learnable weighted summation offer more flexible interactions between the two prompt streams, they do not yield noticeable performance improvements in our setting. One possible explanation is that the two prompts already provide sufficiently complementary information, and more complex merging may introduce redundancy or disrupt this balance.In contrast, attention fusion not only degrades performance, but also incurs substantial memory overhead. This is primarily due to the pairwise similarity computation between all token positions, which generates a large $N \times N$ attention map, where $N$ corresponds to the number of spatial positions in the feature map. For high-resolution inputs, $N$ can be large, making this step particularly memory-intensive.

These findings collectively support our use of simple summation as a practical and efficient fusion strategy in the mixed prompting module. We believe these findings provide a useful reference for future work exploring more adaptive fusion mechanisms in multimodal prompting.

Table 14: Comparison of different prompt fusion strategies in the Mixed Prompting Module.

| Fusion Method | Trainable params | FLOPs | mIoU (%) |
|---|---|---|---|
| Summation | 5.74M | 109.01G | **60.1** |
| Concatenation | 5.83M | 109.09G | 59.4 |
| Learnable Weighted Sum | 5.74M | 109.01G | 59.5 |
| Attention Fusion | 5.84M | 109.09G | 57.0 |

### C.7 Further Validation of Hyperparameter Selection Generalizability

To evaluate the generalizability of our selected settings (e.g., a rank downscale ratio of 4 and 4 subspaces), we conducted additional ablation studies on multiple datasets beyond the NYU Depth v2 dataset originally reported. Specifically, we evaluated the effect of these parameters on MFNet and DELIVER datasets, which encompass RGB-Thermal, RGB-LiDAR, and RGB-Event modalities.

These datasets represent diverse sensing conditions and semantic distributions, offering a broader testing ground for robustness.

Across the diverse datasets evaluated, our selected parameter configuration consistently demonstrated competitive performance and stable optimization behavior. These results indicate that our chosen values strike a favorable balance among model capacity, generalization, and computational efficiency, making them a practical default for a wide range of multimodal segmentation scenarios.

While slight tuning may still benefit extremely domain-shifted settings, our results demonstrate that the chosen configuration is robust and transferable, reducing the burden of per-dataset hyperparameter adjustment in practice.

Table 15: Ablation study on the Ratio (rank downscale ratio).

| Dataset | Modal | Ratio=1 | Ratio=2 | Ratio=4 | Ratio=8 |
|---|---|---|---|---|---|
| NYU Depth v2 | RGB-D | 59.4 | 59.8 | **60.1** | 59.6 |
| MFNet | RGB-T | 58.8 | 59.2 | **60.1** | 59.7 |
| DELIVER | RGB-L | 57.6 | 58.0 | **59.1** | 58.4 |
| DELIVER | RGB-E | 57.2 | 57.8 | **58.0** | 57.8 |

Table 16: Ablation study on the Num (number of subspaces for prompt mixing)

| Dataset | Modal | Num=1 | Num=2 | Num=4 | Num=8 |
|---|---|---|---|---|---|
| NYU Depth v2 | RGB-D | 59.3 | 59.8 | **60.1** | 59.3 |
| MFNet | RGB-T | 58.4 | 58.9 | **60.1** | 59.5 |
| DELIVER | RGB-L | 58.3 | 58.5 | **59.1** | 58.8 |
| DELIVER | RGB-E | 57.1 | 57.6 | **58.0** | 57.5 |

## C.8 Experiments for Different Illumination Conditions

To validate the robustness of our method under varying illumination conditions, we report the performance on the RGB-T MFNet dataset under both daytime and nighttime conditions in Table 17.

On the daytime mIoU, our method achieves an mIoU score of 51.8, ranking second after the best-performing model, CMX (MiT-B4), which scores 52.5. In the nighttime mIoU, a standout performance is achieved, where our model achieves the highest score of 61.0, surpassing all other methods. This demonstrates superior ability of our method to segment objects under nighttime conditions, where challenges such as poor lighting and low visibility are most pronounced.

Table 17: Comparison of RGB-T segmentation performance on the MFNet dataset across daytime and nighttime conditions. The highest mIoU for each condition is highlighted in bold, while the second-best score is underlined.

| Method | Modal | Day | Night |
|---|---|---|---|
| FRRN [59] | RGB | 40.0 | 37.3 |
| DFN [60] | RGB | 38.0 | 42.3 |
| BiSeNet [61] | RGB | 44.8 | 47.7 |
| SegFormer-B2 [34] | RGB | 48.6 | 49.2 |
| SegFormer-B4 [34] | RGB | 49.4 | 52.4 |
| MFNet [10] | RGB-T | 36.1 | 36.8 |
| FuseNet [62] | RGB-T | 41.0 | 43.9 |
| RTFNet [11] | RGB-T | 45.8 | 54.8 |
| FuseSeg [13] | RGB-T | 47.8 | 54.6 |
| GMNet [40] | RGB-T | 49.0 | 57.7 |
| CMX(MiT-B2) [7] | RGB-T | 51.3 | 57.8 |
| CMX(MiT-B4) [7] | RGB-T | **52.5** | 59.4 |
| CMNeXt [8] | RGB-T | 50.5 | _59.8_ |
| Ours | RGB-T | _51.8_ | **61.0** |

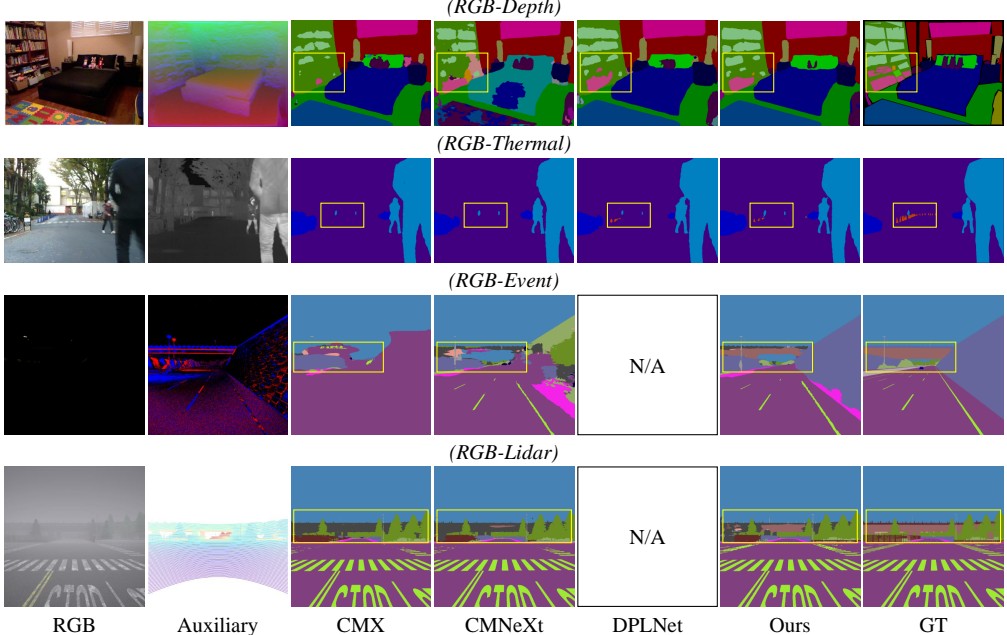

| RGB | Auxiliary | CMX | CMNeXt | DPLNet | Ours | GT |

Figure 5: Visualization of results on the DELIVER dataset under depth, thermal, Event, and Lidar auxiliary modalities. DPLNet is not implemented for RGB-Event and RGB-Lidar data in the original paper.

## C.9 Visualization

To provide a qualitative comparison between our MixPrompt method and existing approaches, we present visualizations of segmentation results from a sample image in each dataset used above. These visualizations are shown in Figure 5.

The comparison illustrates the ability of our model to produce more accurate and refined segmentation boundaries across a variety of auxiliary modalities, including depth, thermal, Event, and Lidar. Overall, the visual comparisons demonstrate the effectiveness of our multimodal fusion framework in generating high-quality segmentation outputs.

## D   Others

### D.1   Social Impact Analysis

Our proposed multimodal fusion method can deploy artificial intelligence more widely in resource constrained environments such as agricultural robots or low-power edge devices, potentially reducing computational costs. However, the dependence on the pre training RGB backbone may inherit the deviation of training data, and the simplified architecture needs strict security verification before being deployed to key applications such as autonomous vehicle. We encourage responsible use and conduct additional fairness and robustness testing.

### D.2   Limitation

Our current work is confined to the image multimodal domain and requires strictly aligned RGB-auxiliary data pairs. The exploration into other multimodal scenarios, such as weakly supervised or unpaired learning tasks is still limited. Addressing these limitations will become the basis of our future research.

