# OpenReview forum: "MixPrompt: Efficient Mixed Prompting for Multimodal Semantic Segmentation"
_NeurIPS.cc/2025/Conference — NeurIPS 2025 poster_

### Official Review · Reviewer_sm6J · 2025-06-14

**Clarity:** 3
**Significance:** 2
**Originality:** 2
**Rating:** 4
**Confidence:** 3

**Summary:**

The paper proposes a framework to incorporate auxiliary modalities, such as depth or lidar, into a pretrained RGB segmentation model to enhance segmentation quality. To achieve better results, it introduces several techniques: initializing the integration module with early layers of a pretrained CNN backbone, and aligning features across multiple low-rank subspaces. Overall, the approach achieves SOTA results on several benchmark datasets.

**Questions:**

1. **Concerns about Paper’s Novelty**

1.1
The concept of a framework that extracts features from depth and then integrates them using a prompt adapter has already been explored in prior work [1].

1.2
Similar initialization techniques have been employed in previous studies to obtain depth features(e.g. making depth "colorful" then using frozen RGB backbone early layers [2]). While the paper conducts more proper research, its novelty appears to be incremental. Could you clarify what is fundamentally new in your method?

2. **Missing Comparison**

Why is there no comparison with OmniVec 2 [3]? Please explain the rationale behind excluding this relevant work.

3. **Parameter Count in Table 1**

Consider adding a column in Table 1 that indicates the number of trainable parameters, especially since your method freezes the backbone and only trains a specific module to demonstrate your efficiency (I saw it later but it will be still valuable in the main table).

4. **RGB-Only Baselines in Table 1**

Please consider including results from RGB-only models in Table 1 to highlight the performance gain of RGB-D fusion. While some modality comparisons are shown later, RGB-D results are missing.

5. **Dataset Selection Justification**

The evaluation focuses on NYU Depth V2 and SUN RGB-D, but widely used datasets such as KITTI and ScanNet are not included. Please explain the rationale behind excluding these datasets from evaluation.

**References**

[1] Dong, Shaohua, Yunhe Feng, Qing Yang, Yan Huang, Dongfang Liu, and Heng Fan. "Efficient multimodal semantic segmentation via dual-prompt learning." In 2024 IEEE/RSJ International Conference on Intelligent Robots and Systems (IROS), pp. 14196-14203. IEEE, 2024.

[2] Gopalapillai, Radhakrishnan, Deepa Gupta, Mohammed Zakariah, and Yousef Ajami Alotaibi. 2021. "Convolution-Based Encoding of Depth Images for Transfer Learning in RGB-D Scene Classification" Sensors 21, no. 23: 7950.

[3] S. Srivastava and G. Sharma, "OmniVec2 - A Novel Transformer Based Network for Large Scale Multimodal and Multitask Learning," 2024 IEEE/CVF Conference on Computer Vision and Pattern Recognition (CVPR), Seattle, WA, USA, 2024, pp. 27402-27414

**Ethical Concerns:**

["NO or VERY MINOR ethics concerns only"]

**Final Justification:**

The authors attempted to address my concerns, but they succeeded only partially.
In the first part, they demonstrated that they generalize previous prompt‑adapter approaches to other modalities.
In the second part, they have not convinced me that their contribution is non‑incremental.
Therefore, I would have preferred to assign a score of 3.5, but since I cannot, I am raising it to 4.

**Limitations:**

yes

**Quality:**

3

**Strengths And Weaknesses:**

**Strengths:**
1. The paper provides well done rationalization for every modification of proposed module, supported by thorough ablation studies.
2. The MixPrompt achieves good performance on proposed benchmark datasets.
3. The authros evaluate the framework across many modalities (e.g. depth, lidar, thermal, event), showing the generalizability of it over modalities.
4. The paper is well-wrtitten and easy to follow.

**Weaknesses:**
1. Incremental novelty of the proposed framework and initialization strategy for auxiliary modalities contributions.

---

> ### Author Rebuttal · Authors · 2025-07-30
>
> **We appreciate reviewer sm6J recognition of our method’s strong performance and broad generalizability as well as the robustness of our experimental evaluation**. Meanwhile, we greatly value the constructive feedback provided and provide detailed clarifications and responses to each point below.
> >**Q1: Concerns about Paper’s Novelty.**
>
> > **Q1.1: The concept of a framework that extracts features from depth and then integrates them using a prompt adapter has already been explored in prior work [1].**
> - We acknowledge that the concept of extracting features from the depth modality and subsequently integrating them using a prompt-based adapter has indeed been explored in prior work, such as DPLNet. However, we note several important limitations in DPLNet design and applicability.
> - DPLNet has primarily been applied to RGB-D and RGB-T settings. While it demonstrates certain improvements on RGB-D datasets, its performance on the RGB-T dataset MFNet is suboptimal, even falling behind the previous state-of-the-art CMNeXt by 0.6 mIoU. This suggests that, **as you mentioned, “it extracts features from depth and then integrates them using a prompt adapter”**, **but such a design may not generalize well, especially when the auxiliary modality differs from depth**. This limitation indicates that DPLNet’s integration strategy is somewhat modality-specific—effective for depth but less adaptable to other auxiliary modalities. In contrast, our proposed method is modality-agnostic: it does not rely on depth-specific feature extraction but instead adopts a generalized prompt-guided fusion mechanism that has been validated across multiple modality combinations, including RGB-Depth, RGB-Thermal, RGB-LiDAR, and RGB-Event.
>
> > **Q1.2: Similar initialization techniques have been employed in previous studies to obtain depth features(e.g. making depth "colorful" then using frozen RGB backbone early layers [2]). While the paper conducts more proper research, its novelty appears to be incremental. Could you clarify what is fundamentally new in your method?**
> - We would like to clarify that the cited Convolution-base Encoding(CBE) method is specifically designed for a two-stream architecture with separate RGB and depth backbones, both of which are fully trainable. This encoding scheme is depth-specific and designed as a replacement for the more complex HHA encoding.
> - In contrast, our approach differs fundamentally in terms of architecture, training strategy, supported modalities, and extensibility. The key distinctions are summarized in *Table 1*.
>   - **Architecture.** We employ a single-stream frozen RGB backbone shared across all modalities, rather than separate RGB and depth streams. Modality-specific adaptation is achieved by injecting learnable prompts into the frozen backbone.
>   - **Training strategy.** As stated in Section 5.2 of the CBE paper, “**all experiments with RGBD CNN were carried out with all layers trainable and randomly initialized weights for the dense layers**.” In contrast, our backbone is completely frozen; only the prompt modules are trained, resulting in substantially higher parameter efficiency.
>   - **Generality.** CBE is depth-specific and cannot be directly extended to other modalities. Our method is modality-agnostic, supporting diverse modalities without redesigning the encoding scheme.
>
>   *Table 1: Comparison between the Convolution-Based Encoding(CBE) method and our MixPrompt.*
>
>   ||**CBE**|**Our MixPrompt**|
>   |:-:|:-:|:-:|
>   |**Task**|Multimodal Classification|Multimodal Semantic Segmentation|
>   |**Architecture**|Two-stream (RGB + depth backbone)| Single-stream (RGB backbone + prompt) |
>   |**Training strategy**|Full parameter training|Backbone freezing|
>   |**Modalities**|Depth-specific only|Depth, Thermal, Event, LiDAR|
>   |**Extensibility**|New modality requires new redesign|New modality requires no redesign|
> ------
>
> >**Q2: Missing Comparison. Why is there no comparison with OmniVec 2 [3]? Please explain the rationale behind excluding this relevant work.**
> - We would like to clarify that OmniVec 2 is fundamentally designed as a general large-scale multimodal and multitask learning framework, whereas our work specifically targets lightweight multimodal semantic segmentation.
> - **OmniVec 2 requires substantial pre-requisite work, including the construction of a large-scale multimodal pretraining dataset and a multi-stage pretraining pipeline** to obtain a strong base model.  In contrast, our MixPrompt does not rely on such extensive pretraining; instead, it achieves competitive results through prompt-based adaptation with a frozen backbone, making it significantly more efficient in both computation and parameter usage.  For clarity, we summarize the major conceptual and practical differences in *Table 2*.
>
>   *Table 2: Comparison between the reviewer-cited OmniVec 2 method and our MixPrompt.*
>
>   ||**OmniVec 2**|**Our MixPrompt**|
>   |:-:|:-:|:-:|
>   |**Task**|General multimodal multitask|Multimodal Semantic Segmentation|
>   |**Architecture**|Two-stream (base BERT) + Transformer|Single-stream (RGB backbone + prompt)|
>   |**Training strategy**|Three-stage pretrain + finetune|Finetune with frozen backbone|
>   |**Trainable Parameters**|Large (value not reported)|small (5.7M)|
>   |**Computation cost**|Large (value not reported)|small (109.0G)|
> - Despite the above differences, we still report a quantitative comparison with OmniVec 2 and its predecessor OmniVec *[4]* on the NYU dataset.  Results show that our method, with significantly fewer parameters and much lower computational cost, surpasses OmniVec's finetuned performance on NYU.  While OmniVec 2 still achieves higher accuracy, it comes at the cost of extremely large-scale pretraining and computational demands, which are not aligned with the scope and design philosophy of our work.
>
>   *Table 3: Quantitative comparison on the NYU dataset. Pre: pretrained only; FT: fine‑tuned on NYU.*
>
>   |Method|Re‑Pretrain|mIoU (%)|
>   |:-:|:-:|:-:|
>   |OmniVec (Pre)|✔|58.6|
>   |OmniVec (FT)|✔|60.8|
>   |OmniVec 2 (FT)|✔|63.6|
>   |**MixPrompt (Ours)**|✘|61.2|
> ------
>
> >**Q3: Parameter Count in Table 1. Consider adding a column in Table 1 that indicates the number of trainable parameters, especially since your method freezes the backbone and only trains a specific module to demonstrate your efficiency (I saw it later but it will be still valuable in the main table).**
>
> - We agree that explicitly reporting the number of trainable parameters in Table 1 would provide a clearer illustration of the efficiency of our method and we provide a sample table as follows due to space limitations. **We commit to revising Table 1 in the final version to include trainable parameter counts for all methods**.
>
>   *Table 4: Sample table.*
>
>   |Model|Backbone|Params(Trainable)|NYU: Input/FLOPs/mIoU|SUN: Input/FLOPs/mIoU|
>   |:-:|:-:|:-:|:-:|:-:|
>   |ACNet|ResNet50|116.6M(116.6M)|480×640/126.7G/48.3%|530×730/163.9G/48.1%|
> ------
> >**Q4: RGB-Only Baselines in Table 1. Please consider including results from RGB-only models in Table 1 to highlight the performance gain of RGB-D fusion. While some modality comparisons are shown later, RGB-D results are missing.**
>
> - Following your advice, we have conducted additional experiments using RGB-only baselines. The results are summarized in *Table 5*.  **We will include these RGB-only results in Table 1 of the final version**.
>
>   *Table 5: RGB-only models on the RGB-D dataset.*
>
>   |Model|Backbone|Params|NYU: Input/FLOPs/mIoU|SUN: Input/FLOPs/mIoU|
>   |:-:|:-:|:-:|:-:|:-:|
>   |SegFormer|MiT-B4|62.44M|480×640/74.7G/52.3%|530×730/96.3G/49.1%|
> ------
>
> >**Q5: Dataset Selection Justification. The evaluation focuses on NYU Depth V2 and SUN RGB-D, but widely used datasets such as KITTI and ScanNet are not included. Please explain the rationale behind excluding these datasets from evaluation.**
>
> - We would like to clarify that NYU Depth V2 and SUN RGB-D are the two most widely used benchmarks for RGB-D semantic segmentation, and **we follow prior works *[1,2,3,4,5,6,8]* in conducting evaluation on these datasets**.
> - We would like to further elaborate on our considerations and analysis regarding the datasets mentioned by the reviewer:
>   - **KITTI.**  KITTI Vision Benchmark *[7]* is primarily an object detection dataset. Although the official KITTI Semantic Segmentation Evaluation *[8]* (containing 200 training and 200 testing RGB images) was later released, the lack of auxiliary modalities has limited its use in multimodal semantic segmentation research.
>   - **ScanNet.** ScanNet *[9]* is a large-scale RGB-D video dataset containing over 2.5 million views, annotated with 3D camera poses, surface reconstructions, and instance-level semantic segmentations.  We acknowledge ScanNet as a valuable indoor scene benchmark and have submitted the required agreement form but have not yet received access approval. We are committed to including ScanNet evaluation results in the final paper.
>
> **Reference**
>
> *[1] Efficient multimodal semantic segmentation via dual-prompt learning. IROS, 2024.*
>
> *[2] Convolution-Based Encoding of Depth Images for Transfer Learning in RGB-D Scene Classification. Sensors, 2021*
>
> *[3] OmniVec2 - A Novel Transformer Based Network for Large Scale Multimodal and Multitask Learning. CVPR, 2024*
>
> *[4] Omnivec: Learning robust representations with cross modal sharing. WACV, 2024.*
>
> *[5] Cmx: Cross-modal fusion for rgb-x semantic segmentation with transformers. IEEE T-ITS, 2023*
>
> *[6] Delivering arbitrary-modal semantic segmentation. CVPR, 2023*
>
> *[7] Are we ready for autonomous driving? the kitti vision benchmark suite. CVPR, 2012*
>
> *[8] Augmented reality meets computer vision: Efficient data generation for urban driving scenes. IJCV, 2018*
>
> *[9] Scannet: Richly-annotated 3d reconstructions of indoor scenes. CVPR, 2017*

---

> > ### Comment · Reviewer_sm6J · 2025-08-05
> >
> > I am sorry for the delay.
> > Thank you for your detailed response and for addressing my concerns. I appreciate the clarifications provided, which have resolved most of my concerns.
> > However, I still have a concern regarding the contribution"reusing early layers of a pretrained RGB backbone is an effective embedding strategy for auxiliary modalities". It has been already explored in previous works and your contribution still looks incremental for me.
> > I will carefully consider your response when finalizing my score and justification.

---

> ### Author Response · Authors · 2025-08-05
> **Hope to Get Your Reply**
>
> Dear Reviewer sm6J,
>
> As the deadline is nearing, we wanted to gently follow up on our recent submission. We have meticulously addressed each of your suggestions one by one and incorporated these feedbacks into the revised version. During the rebuttal period, We have discussed and clarified the distinctions from prior works, including DPLNet and CBE. Furthermore, we **supplemented our response with a comparative analysis against** **OmniVec 2** and provided justification for our dataset selection. We also **carried out supplementary experiments** to assess the performance of the RGB single modality on NYU and SUNRGBD. We hope that these efforts will alleviate your concerns regarding Aligner. Your feedback is highly valuable to us, and we would appreciate any updates or further guidance you might have regarding our revisions and responses.
>
> Thank you for your time and consideration.

---

> ### Comment · Area_Chair_bevN · 2025-08-05
> **Update after rebuttal**
>
> Dear Reviewer,
>
> The authors’ rebuttal has been posted. Please check the authors’ feedback, evaluate how it addresses the concerns you raised, and post any follow-up questions to engage the authors in a discussion. Please do this ASAP.
>
> Thanks.
>
> AC

---

> ### Author Response · Authors · 2025-08-06
> **(Round 2) Response to Reviewer sm6J**
>
> Dear Reviewer sm6J,
>
> We are deeply grateful for your support of Mixprompt, and it has been highly encouraging for us to address your concerns in the rebuttal. We fully understand your concern regarding the contribution *“reusing early layers of a pretrained RGB backbone as an effective embedding strategy for auxiliary modalities.”*
>
> In reference to the work you mentioned *[1]*, we have **conducted additional experiments** by replacing our pretrained RGB backbone with their proposed  convolutional encoder on both RGB-D and RGB-T datasets. The results are summarized below.
>
> *Table 1: Comparison of different encoding method in the Mixed Prompting Module*
>
> |Encoder Method|NYUDepth (RGB-D)|MFNet (RGB-T)|
> | :-: | :-: | :-: |
> | Convolution-Based Encoding *[1]* |       58.7       |     56.8      |
> |               Ours               |     **61.2**     |   **60.1**    |
>
> As shown in the *Table 1*, the previous depth-image encoding method does not adapt well to the prompting framework. It may provide a suboptimal prompt initialization, leading to a performance drop of about 2.5% mIoU compared to our trainable backbone on RGB-D datasets. Moreover, results on the RGB-T dataset indicate that it does not generalize effectively to arbitrary modality encoders, which limits its scalability. **Overall, these experiments provide meaningful evidence for our design choice, and the comparative analysis will be included in the final version of the paper.**
>
> Would it be possible for you to consider raising your score after we have addressed your concerns? If there are any additional points you believe we should incorporate, please let us know, and we will promptly reply to you and try our best to incorporate them into the revised version of the paper.
>
> With best regards!
>
> *[1] Gopalapillai, Radhakrishnan, Deepa Gupta, Mohammed Zakariah, and Yousef Ajami Alotaibi. 2021. "Convolution-Based Encoding of Depth Images for Transfer Learning in RGB-D Scene Classification" Sensors 21, no. 23: 7950.*

---

> > ### Comment · Reviewer_sm6J · 2025-08-06
> >
> > Thank you for conducting the additional experiment. I will take it into consideration when submitting my score.

---

> > > ### Author Response · Authors · 2025-08-07
> > > **Thank you to your recognition and encouragement!**
> > >
> > > Dear Reviewer sm6J,
> > >
> > > Thank you once again for your time and consideration. I hope the additional experiment results help clarify our contributions and are favorably received. I look forward to your final decision.

---

### Official Review · Reviewer_LKhh · 2025-06-30

**Clarity:** 4
**Significance:** 3
**Originality:** 3
**Rating:** 4
**Confidence:** 4

**Summary:**

This paper proposes a MixPrompt framework to multi-modal semantic segmentation task, which adopts prompt tuning to integrate information from other modalities into the processing of RGB. To further improve the performance and efficiency, the authors propose Multi-subspace prompting method to project RGB features into multiple subspaces and mix them with the prompt features from the auxiliary branch. The authors conduct extensive experiments on NYU Depth V2, SUN-RGBD, MFNet, and DELIVER datasets to validate the effectiveness of the proposed method.

**Questions:**

1. Providing more visual observations of the multi-subspace distributions would better illustrate the effect of the proposed method.
2. It would be better to clearly display and distinguish different experimental settings in the table.
3. Can the authors demonstrate more experiments based on the ImageNet pre-trained backbone to verify the benefits of pre-training on ADE20K, even unfreeze all parameters?

**Ethical Concerns:**

["NO or VERY MINOR ethics concerns only"]

**Final Justification:**

The author's response addressed most of the questions, and the method is effective. However, I am still concerned about the consistency comparison of the methods, and the incremental innovation. I would like to raise the rating to 'Borderline', but there is no such level, I raised it to 'Borderline accept'.

**Limitations:**

yes

**Quality:**

3

**Strengths And Weaknesses:**

Strengths:
1. The proposed framework seems reasonable. The prompt learning method based on the pre-trained model has indeed greatly improved the performance and training efficiency on many tasks.
2. The experimental results are good on several benchmarks.

Weaknesses:
1. The motivation for proposing Multi-subspace prompting is unclear. The authors did not use theoretical derivation or experimental observation to derive the reason why RGB or other modality features need to be projected to multiple subspaces, which requires a deeper analysis.
2. Regarding the method, the method proposed in this paper is more like an incremental improvement of the prompt tuning method proposed by DPLNet, which constructs a multi subspace mapping layer based on DPLNet. Therefore, the contribution is somewhat insufficient.
3. For the experiments, the method in this paper is based on the backbone pre-trained on ADE20K, which is relatively unfair to compare with the previous method pre-trained on ImageNet, like DFormer. I understand that different method paradigms have different starting points, but the author should clearly indicate it in the comparison table, which is more friendly to other readers.

---

> ### Author Rebuttal · Authors · 2025-07-29
>
> **We greatly appreciate reviewer LKhh acknowledgement of our method’s strong performance, efficient training, and comprehensive experimental validation**. At the same time, we highly appreciate the thoughtful suggestions offered and present comprehensive explanations and clarifications for each concern below.
>
> >**Q1: Providing more visual observations of the multi-subspace distributions would better illustrate the effect of the proposed method.**
>
> - While the rebuttal format restricts the inclusion of visualizations, **we conducted additional quantitative analysis to characterize the diversity among the learned subspaces through detailed tabular statistics**. Following the hierarchical encoder architecture of SegFormer, our backbone extracts four-stage multi-scale features with progressively lower spatial resolutions, denoted as d1–d4 (from highest to lowest resolution). For each stage, we compute the average pairwise cosine similarity between projected RGB subspace features, where cosine similarity values range from -1 (completely dissimilar) to 1 (identical). Lower (more negative) similarity values indicate greater diversity across subspaces. The Avg.Sim (overall) column summarizes the mean similarity across all four stages, providing an overall measure of inter-subspace diversity.
> - As reported in *Table 1*, we use the RGB-D inputs from the NYU Depth V2 test set to evaluate how varying the number of subspaces (n) affects the diversity of learned subspace features. Our analysis reveals that, within the same number of subspaces (n), similarity values become increasingly negative from high-resolution (d1) to low-resolution (d4) features. This is because high-resolution features predominantly capture modality-agnostic local patterns (such as edges and textures) which remain relatively consistent across subspaces, whereas lower-resolution features encode more abstract, global, and cross-modal semantics. Consequently, our multi-subspace prompt mixing more effectively disentangles these higher-level representations, resulting in greater divergence at coarser scales.
> - In addition, increasing n from 1 to 4 substantially reduces overall similarity (from -0.0514 to -0.1922), reflecting enhanced structural diversity and correlating with the best segmentation performance (mIoU of 60.1\%). However, further increasing n to 8 causes the similarity magnitude to shrink (from -0.1922 to -0.0169), indicating over-fragmentation that diminishes subspace diversity and aligns with the observed performance degradation.
>
>   *Table 1: Average pairwise cosine similarity between RGB subspace features across different numbers of subspaces ($n$) and corresponding segmentation performance.*
>
>   |n|Avg.Sim (d1)|Avg.Sim (d2)|Avg.Sim (d3)|Avg.Sim (d4)|Avg.Sim (overall)|mIoU (%)|
>   |:-:|:-:|:-:|:-:|:-:|:-:|:-:|
>   |1|N/A|N/A|N/A|N/A|N/A|59.3|
>   |2|-0.0338|-0.0355|-0.0508|-0.0853|-0.0514|59.8|
>   |4|-0.1434|-0.1715|-0.2031|-0.2506|-0.1922|60.1|
>   |8|-0.0135|-0.0046|-0.0175|-0.0321|-0.0169|59.6|
>
> - These quantitative results indirectly validate the core motivation behind our multi-subspace design: by introducing structural diversity and encouraging disentangled prompt composition, it effectively improves segmentation outcomes. We acknowledge that providing more visual observations of the multi-subspace distributions would offer an even clearer and more intuitive demonstration of the proposed method’s effectiveness. **Due to rebuttal format constraints, we hope this detailed statistical characterization serves as a compelling alternative and we plan to include comprehensive visualizations in the final version of the paper**.
>
> ------
>
> >**Q2: It would be better to clearly display and distinguish different experimental settings in the table.**
>
> - Thank you for your valuable suggestion. Although we have provided detailed configuration information in the “Additional Experiment Details” section of the supplementary materials, we agree that it is not very intuitive for readers to quickly grasp. Therefore, we will provide a detailed and comprehensive listing of all relevant configurations. Due to space limitations, we present here a simplified example table as *Table 2*. **We commit to including in the final version a comprehensive and clearly organized table summarizing all experimental settings across datasets**.
>
>   *Table 2: Training settings sample table.*
>
>   |Dataset|input size|optimizer|learning rate|weight decay|batch size|epochs|optimizer momentum|warmup epochs|warmup schedule|learning rate schedule|
>   |:-:|:-:|:-:|:-:|:-:|:-:|:-:|:-:|:-:|:-:|:-:|
>   |SUN-RGBD|530×730|AdamW|5e-3|0.01|4|100|β1, β2=0.9, 0.999|5|linear|polynomial decay|
>
> ------
>
> >**Q3: Can the authors demonstrate more experiments based on the ImageNet pre-trained backbone to verify the benefits of pre-training on ADE20K, even unfreeze all parameters?**
> - We agree that our method uses a backbone pre-trained on ADE20K, whereas some prior works such as DFormer are based on ImageNet pre-training. **To clarify, DFormer does not simply use ImageNet pre-trained weights. It begins by creating RGB-Depth image pairs derived from the ImageNet dataset through a detailed and involved preprocessing step, and then conducts an additional pre-training stage on these pairs, adding unnecessary complexity to the workflow**. Different methods have their own variations in how pretrained weights are loaded and utilized, leading to discrepancies in model initialization. We commit to fully considering these differences and clearly indicating them in the comparison tables and the final version of the paper for a more reader-friendly and transparent evaluation.
> - Following your suggestion, we conducted additional experiments using the ImageNet-pretrained backbone under both frozen and unfrozen settings. The results are summarized in the table below.
>
>   *Table 3: Comparison of different backbone pretraining settings.*
>
>   | Backbone Pretraining | Backbone Frozen | Trainable Params |   mIoU   |
>   |:-:|:-:|:-:|:-:|
>   |MiT-B5 (ImageNet)|✓|5.7M|50.4|
>   |MiT-B5 (ImageNet)|✗|87.2M|56.3|
>   |MiT-B5 (ADE20K)|✓|5.7M|**61.2**|
>
> - As observed, even when fully fine-tuning the ImageNet-pretrained backbone, the performance is still noticeably lower than using the ADE20K-pretrained parameter with a frozen backbone. This is likely because ImageNet pre-training focuses on image classification, which does not equip the model with rich segmentation-related semantic knowledge. Consequently, the prompt learning module struggles to extract effective task-specific information from such backbones. In contrast, the ADE20K-pretrained backbone already encodes strong segmentation features, enabling the prompt module to leverage more informative representations and thus achieve better performance.
> - We will include these experimental details and analyses explicitly in the final manuscript to address the reviewer’s concerns and ensure a fair and comprehensive comparison.
>
> ------
>
> >**Additional Response about DPLNet**
>
> - We are grateful for the reviewer’s insightful comment on the relationship between our method and DPLNet. While both approaches are indeed inspired by prompt learning, they differ fundamentally in their design and capabilities across several key dimensions.
>   - **Prompt adaptation mechanism.** DPLNet adapts the query features in a frozen backbone by inserting a set of learnable tokens into the attention computation. However, such token-based adaptation has inherent limitations: (i) the adaptation happens indirectly via attention weights, which may weaken the prompt’s ability to inject explicit task- or modality-specific cues; (ii) the fixed set of tokens cannot fully capture the diverse and complementary information required across different modalities.  In contrast, our multi-subspace prompt mixing approach directly projects prompts into multiple learnable subspaces and fuses them without modifying the attention queries inside the backbone. This design better aligns with the original motivation of prompt-based architectures—decoupling modality adaptation from backbone internals—while enabling richer and more flexible cross-modal alignment.
>   - **Generalization capability.** DPLNet’s applicability is restricted to RGB-D and RGB-T settings, and its generalization is limited. Remarkably, on the MFNet dataset (RGB-T), DPLNet performs worse than CMNeXt (-0.6 mIoU), suggesting that its learned prompts may be overfitted to specific modality combinations rather than capturing modality-agnostic patterns.  In contrast, MixPrompt achieves state-of-the-art results across five different modality settings, including RGB-D, RGB-T, RGB-Radar, RGB-Event, and RGB-LiDAR, without requiring architectural modifications or retraining. This strong cross-modal performance indicates that our design is a step toward a more general and scalable multimodal segmentation framework.
> - In summary, the differences in prompt adaptation strategy and generalization capability make MixPrompt a distinct and more versatile formulation compared to DPLNet, addressing the limitations of token-based adaptation while demonstrating superior applicability across diverse multimodal scenarios.

---

> > ### Comment · Reviewer_LKhh · 2025-08-06
> >
> > Thank you for the author's thoughtful and detailed response. I appreciate the efforts and experiments the author has made in the response, which has well addressed most of my questions.
> >
> > My concerns about the fairness of the experimental setup still remain. As the author shows in Table 3, pre-training on ADE20k does bring significant performance improvements. This part of the improvement comes from the pre-training task rather than the method itself. This is why I hope author will show these different settings in the comparative experiments.
> >
> > Meanwhile, the issue of incremental innovation has also been mentioned by other reviewers.
> >
> > I hope the above can help you improve your work. I will think carefully and give my final rating.

---

> ### Author Response · Authors · 2025-08-05
> **Hope to Get Your Reply**
>
> Dear Reviewer LKhh,
>
> As the deadline approaches, we wanted to kindly follow up on our recent submission. We have carefully addressed each of your suggestions and incorporated the feedback into the revised version. During the rebuttal period, we have **supplemented our response with both theoretical analysis and experimental evidence** to better illustrate the effect of the proposing Multi-subspace prompting method. Following your suggestion, we also **conducted experiments with different pre-trained backbones**, which confirmed the benefits of pre-training on ADE20K. Furthermore, we explicitly **clarified the fundamental differences between MixPrompt and DPLNet** regarding prompt adaptation and generalization capability. Your feedback has been invaluable, and we would greatly appreciate any updates or further guidance you may have regarding our revisions and responses.
>
> Thank you for your time and consideration.

---

> ### Comment · Area_Chair_bevN · 2025-08-05
> **Update after rebuttal**
>
> Dear Reviewer,
>
> The authors’ rebuttal has been posted. Please check the authors’ feedback, evaluate how it addresses the concerns you raised, and post any follow-up questions to engage the authors in a discussion. Please do this ASAP.
>
> Thanks.
>
> AC

---

> ### Author Response · Authors · 2025-08-07
> **(Round 2) Response to Reviewer LKhh**
>
> Dear Reviewer LKhh,
>
> Thank you for your response. We are greatly encouraged that `most of your concerns have been addressed`. The significant efforts we put in during the rebuttal period have proven to be worthwhile.
>
> > the fairness of the experimental setup
>
> We fully understand your concern regarding the *fairness of the experimental setup.* To alleviate your concern, we have **conducted additional experiments** where we initialized several prior multi-modal approaches (e.g., CMX and CMNeXt) with ADE20K-pretrained weights, followed by training on the NYUDepth dataset.  The results are summarized below.
>
> *Table 1: Comparison of different method with different backbone pretraining settings.*
>
> |Method|Backbone Pretraining|**Trainable Params**|**mIoU**|
> |:-:| :-: |:-:|:-:|
> |Segformer|   MiT-B5 (ADE20K)|82.7M|54.7|
> |CMX|MiT-B5 (ImageNet)|181.1M|56.9|
> |CMX|MiT-B5 (ADE20K)|181.1M|58.1|
> |CMNeXt|MiT-B4 (ImageNet) |119.6M|56.9|
> |CMNeXt|MiT-B4 (ADE20K)|119.6M|58.3|
> |Ours|MiT-B5 (ADE20K)|**5.7M**|**61.2**|
>
> As shown in the *Table 1*, We observe that CMX and CMNeXt do benefit from using the same strong baseline as ours, resulting in limited performance gains. **However, the gap between their results and ours remains substantial (e.g., a 2.9–3.1 mIoU difference on the NYUDepth dataset).** This comparison under the same pretrained setting demonstrates the effectiveness of our method over other multi-modal fusion approaches. Notably, our model achieves this with only 5.7M trainable parameters—significantly fewer than the alternatives. Additionally, we conducted experiments where the single-modality RGB backbone (SegFormer) was also initialized with ADE20K-pretrained weights. **The results further indicate that the performance gains achieved by our method go well beyond what can be attributed to pretraining alone, highlighting the strength of our fusion strategy**.
>
> **Overall, these experiments provide meaningful evidence for our fairness of the experimental setup, and the comparative analysis will be included in the final version of the paper.**
>
> > the issue of incremental innovation
>
> We fully understand your concern regarding the novelty of our approach, and we appreciate the opportunity to further clarify the conceptual distinctions from prior work, particularly DPLNet. While both DPLNet and MixPrompt are inspired by the idea of prompt-driven multimodal integration, their design principles and implementation paradigms are fundamentally different.
>
> **DPLNet injects learnable prompts directly into the attention computation of each backbone block**, requiring architectural modifications at every layer. It freezes the modality-specific branches and introduces prompts as additional key-value inputs within each attention module to enable cross-modal interactions. **This approach tightly couples the prompt design with the internal computation of the backbone, increasing complexity and limiting generalizability** (DPLNet underperforms on the MFNet RGB-T dataset despite the relatively mild conditions).
>
> **In contrast, MixPrompt introduces a lightweight and modular prompting mechanism that operates externally to the backbone attention layers.** Rather than modifying the internal architecture, we leverage a pretrained RGB backbone as a shared encoder across modalities and introduce learnable prompts at selected locations to guide the representation of auxiliary modalities. This enables effective cross-modal adaptation without altering the backbone structure or requiring modality-specific branches.
>
> Moreover, **our multi-subspace prompting strategy provides a novel mechanism for aligning and injecting auxiliary modality information**. By decomposing the feature space into distinct subspaces and associating prompts accordingly, MixPrompt captures fine-grained modality cues while maintaining model compactness and deployment flexibility. This modular design allows us to support new modalities (e.g., radar, event) without retraining or redesigning the network—an advantage that DPLNet does not offer.
>
> In summary, MixPrompt represents more than an incremental improvement. It introduces a shift in how prompting is applied—from internal attention modification to external modular guidance—resulting in a framework that is not only efficient and robust, but also broadly generalizable. We hope this explanation helps clarify the novelty and practical value of our approach.
>
> We sincerely hope that you will consider raising the assessment of Mixpromt. If there are any additional points you believe we should incorporate, please let us know, and we will promptly reply to you and try our best to incorporate them into the revised version of the paper.
>
> With best regards!

---

> > ### Comment · Reviewer_LKhh · 2025-08-07
> >
> > Thank you for the author's detailed response, I will consider the above content as the basis for my final rating.

---

> > > ### Author Response · Authors · 2025-08-07
> > > **Thank you to your recognition and encouragement!**
> > >
> > > Dear Reviewer LKhh,
> > >
> > > I would like to express my gratitude once again for your time and thoughtful consideration. It is my hope that the additional experimental results offer meaningful insight and are taken into account favorably in your assessment. I look forward to your final decision.

---

### Official Review · Reviewer_cD6A · 2025-07-02

**Clarity:** 3
**Significance:** 3
**Originality:** 2
**Rating:** 4
**Confidence:** 5

**Summary:**

This paper presents MixPrompt, a prompting-based framework for multimodal semantic segmentation that integrates auxiliary modalities into a pretrained RGB segmentation model. The method introduces a lightweight prompting module and a multi-subspace alignment strategy to effectively adapt the model to multimodal inputs without retraining the entire backbone. The proposed approach is evaluated on multiple benchmark datasets, demonstrating competitive performance in multimodal segmentation tasks.

**Questions:**

- Given that multimodal semantic segmentation (e.g., RGB-Thermal, RGB-Depth) is a mature area, what do you see as the practical advantage of MixPrompt over prior methods? Does it offer clear benefits in efficiency, robustness, or generalizability that would motivate its adoption?
- The paper claims that multi-subspace prompting enhances information utilization. Can you provide empirical or theoretical support for this claim?
- You performed ablation studies on a single dataset to select the rank downscale ratio and the number of subspaces for prompt mixing. Can you clarify whether the selected values (e.g., 4) generalize well across other datasets, or whether they require dataset-specific tuning? This would help assess the robustness and usability of your method in broader settings.
- The reported performance gains are relatively small (e.g., 0.5–1.0% mIoU in some cases), and the visual results in Figure 5 do not clearly indicate a qualitative advantage. Could you provide further evidence to justify the practical impact of MixPrompt?

**Ethical Concerns:**

["NO or VERY MINOR ethics concerns only"]

**Final Justification:**

The rebuttal provided additional experiments and clarifications, addressing my questions on efficiency, robustness, and generalizability. I recognize the practical value of the lightweight, flexible design. However, my concern about the limited conceptual novelty, especially the incremental differences from prior prompting-based methods such as DPLNet, remains. Given the technical soundness and broad applicability, I maintain a borderline rating.

**Limitations:**

yes

**Quality:**

3

**Strengths And Weaknesses:**

**Strengths**
- The paper proposes a simple yet effective prompting-based approach that can be efficiently integrated into existing pretrained RGB segmentation models.
- The method demonstrates strong performance across multiple benchmark datasets for multimodal semantic segmentation, validating its general applicability.
- The framework offers a computationally efficient solution that avoids retraining large backbones, which is especially appealing for real-world deployment scenarios.

**Weaknesses**
- The core concept of using a pretrained RGB model while injecting auxiliary modalities through prompts was already adopted in prior work such as [14]. As a result, the originality of MixPrompt is limited to the specific design of its prompting module.
- Multimodal semantic segmentation, particularly in RGB-Thermal or RGB-Depth settings, is a well-established field with strong existing baselines. The paper does not clearly differentiate MixPrompt from prior methods in terms of either methodology or practical impact.
- The paper does not sufficiently clarify why the proposed method works. For instance, the effectiveness of multi-subspace prompting is attributed to improved information exploitation through multiple down- and up-projection pairs, but this claim lacks theoretical justification.
- In the ablation study, key design choices, such as setting the rank downscale ratio and the number of subspaces for prompt mixing to four, are not well motivated. It remains unclear whether these hyperparameters are dataset-specific or reflect more general design principles.
- In several experiments, the performance improvements over state-of-the-art methods are relatively modest (e.g., 0.5–1.0% mIoU), raising concerns about the practical significance of the proposed approach. Additionally, the qualitative results presented in Figure 5 do not clearly demonstrate a noticeable visual improvement, making it difficult to assess the perceptual advantage of MixPrompt.

---

> ### Author Rebuttal · Authors · 2025-07-28
>
> **We are pleased that the reviewer cD6A found our method to be simple yet effective, easy to integrate, and highly generalizable, which makes it particularly appealing for real-world deployment scenarios**. Most of the concerns raised seem to stem from misunderstandings of our work, which we address point-by-point below.
>
> ------
>
> >**Q1: Given that multimodal semantic segmentation (e.g., RGB-Thermal, RGB-Depth) is a mature area, what do you see as the practical advantage of MixPrompt over prior methods? Does it offer clear benefits in efficiency, robustness, or generalizability that would motivate its adoption?**
> - As noted in the *Introduction* and *Related Works*, compared to existing multimodal segmentation methods that often rely on modality-specific branches and heavy fusion modules, MixPrompt introduces clear advantages in terms of efficiency, robustness, and generalizability.
>   - **Efficiency.**  MixPrompt achieves superior efficiency with only **5.7M trainable parameters** required to adapt to downstream tasks, which is significantly lower than prior methods such as CMX-B5 (181.1M), DFormer (39.0M), and DPLNet (7.15M). Despite its lightweight nature, MixPrompt outperforms all of these methods in segmentation accuracy, demonstrating that our prompt-based design achieves a better trade-off between performance and parameter overhead.
>   - **Robustness.**  The DELIVER dataset includes diverse challenging conditions such as cloudy, foggy, nighttime, and rainy environments, as well as sensor degradations including Motion Blur (MB), Over-Exposure (OE), LiDAR-Jitter (LJ), Event Low-resolution (EL) and Under-Exposure (UE). As shown in **Table 3 of our paper**, under these adverse conditions and corner cases, MixPrompt consistently outperforms previous methods such as CMX and CMNeXt. In addition, on the MFNet dataset, which contains both daytime and nighttime scenes, DPLNet fails to surpass earlier baselines like CMX and CMNeXt under relatively mild environmental disturbances. In contrast, MixPrompt achieves state-of-the-art performance, highlighting its robustness in real-world multimodal scenarios.
>   - **(3) Generalizability.**  **As kindly noted in your review, “the method demonstrates state-of-the-art results across multiple benchmark datasets for multimodal semantic segmentation, validating its general applicability.”** Unlike prior methods (DPLNet) that are typically restricted to RGB-D or RGB-T settings, MixPrompt generalizes well to additional modalities such as radar and event data without requiring architectural modifications or retraining, thanks to its modular and flexible prompt design.
> - In summary, MixPrompt achieves better performance than prior work such as DPLNet in terms of efficiency, generality, and robustness, while being theoretically motivated and empirically validated under more challenging multimodal segmentation scenarios.
>
> ------
>
> >**Q2: The paper claims that multi-subspace prompting enhances information utilization. Can you provide empirical or theoretical support for this claim?**
> - As illustrated in **Algorithm 1 of our supplementary material**, our approach decomposes the high-dimensional features into *n* subspaces and performs linear mixing independently within each. This subspace-wise interaction scheme encourages diverse and disentangled feature combinations, which is beneficial for multi-modal fusion. **From the perspective of information theory, such factorized subspace interactions can be interpreted as promoting conditional independence and reducing redundancy, thereby potentially increasing the mutual information between the resulting prompt and the fused input modalities**.
> - In addition, we present additional quantitative analyses with detailed tabular statistics to assess the diversity of the learned subspaces. Our backbone outputs four-stage multi‑scale features (d1–d4, high→low resolution). For each stage, we compute the average pairwise cosine similarity between projected RGB subspace features (−1 to 1, lower means greater diversity).
> - As reported in *Table 1*, increasing $n$ from 1 to 4 markedly lowers overall similarity (−0.0514→−0.1922), boosting structural diversity and yielding the best mIoU (60.1%). Further raising \(n\) to 8 reverses this trend (−0.1922→−0.0169), suggesting over‑fragmentation that reduces diversity and degrades performance. **These quantitative results validate the core motivation behind our multi-subspace design: by introducing structural diversity and encouraging disentangled prompt composition, it effectively improves segmentation outcomes**.
>
>   *Table 1: Average pairwise cosine similarity between subspace features across different numbers of subspaces ($n$) .*
>   |n|Avg.Sim (d1)|Avg.Sim (d2)|Avg.Sim (d3)|Avg.Sim (d4)|Avg.Sim (overall)|mIoU (%)|
>   |:-:|:-:|:-:|:-:|:-:|:-:|:-:|
>   |1|N/A|N/A|N/A|N/A|N/A|59.3|
>   |2|-0.0338|-0.0355|-0.0508|-0.0853|-0.0514|59.8|
>   |4|-0.1434|-0.1715|-0.2031|-0.2506|-0.1922|60.1|
>   |8|-0.0135|-0.0046|-0.0175|-0.0321|-0.0169|59.6|
>
> - **We commit to including the above experiments along with more comprehensive visualizations in the final version of the paper to better support our method**.
>
> ------
>
> >**Q3: You performed ablation studies on a single dataset to select the rank downscale ratio and the number of subspaces for prompt mixing. Can you clarify whether the selected values (e.g., 4) generalize well across other datasets, or whether they require dataset-specific tuning? This would help assess the robustness and usability of your method in broader settings.**
> - To evaluate the generalizability of our selected settings (e.g., a rank downscale ratio of 4 and 4 subspaces), we conducted additional ablation studies on multiple datasets beyond the NYU Depth v2 dataset originally reported. Specifically, we evaluated the effect of these parameters on MFNet and DELIVER datasets, which encompass *RGB-Thermal*, *RGB-LiDAR*, and *RGB-Event* modalities. These datasets represent diverse sensing conditions and semantic distributions, offering a broader testing ground for robustness.
> - Across the diverse datasets evaluated, our selected parameter configuration consistently demonstrated competitive performance and stable optimization behavior. These results indicate that our chosen values strike a favorable balance among *model capacity*, *generalization*, and *computational efficiency*, making them a practical default for a wide range of multimodal segmentation scenarios.
>
>   *Table 2: Ablation study on the Ratio (rank downscale ratio).*
>   |Dataset|Modal|Ratio=1|Ratio=2|Ratio=4|Ratio=8|
>   |:-:|:-:|:-:|:-:|:-:|:-:|
>   |NYU Depth v2|RGB-D|59.4|59.8|**60.1**|59.6|
>   |MFNet|RGB-T|58.8|59.2|**60.1**|59.7|
>   |DELIVER|RGB-L|57.6|58.0|**59.1**|58.4|
>   |DELIVER|RGB-E|57.2|57.8|**58.0**|57.8|
>
>   *Table 3:  Ablation study on the Num (number of subspaces for prompt mixing)*
>   |Dataset|Modal|Num=1|Num=2|Num=4|Num=8|
>   |:-:|:-:|:-:|:-:|:-:|:-:|
>   |NYU Depth v2|RGB-D|59.3|59.8|**60.1**|59.3|
>   |MFNet|RGB-T|58.4|58.9|**60.1**|59.5|
>   |DELIVER|RGB-L|58.3|58.5|**59.1**|58.8|
>   |DELIVER|RGB-E|57.1|57.6|**58.0**|57.5|
>
> - While slight tuning may still benefit extremely domain-shifted settings, our results demonstrate that the chosen configuration is **robust and transferable**, reducing the burden of per-dataset hyperparameter adjustment in practice.
> ------
>
> >**Q4: The reported performance gains are relatively small (e.g., 0.5–1.0% mIoU in some cases), and the visual results in Figure 5 do not clearly indicate a qualitative advantage. Could you provide further evidence to justify the practical impact of MixPrompt?**
>
> - We would like to highlight that **MixPrompt is designed with parameter efficiency as a core motivation**: during training, *only the prompting module and the segmentation head are updated*, while the RGB backbone remains frozen. This significantly reduces the number of trainable parameters and computational cost. Despite this constraint, MixPrompt consistently improves performance across diverse datasets
>
> - While the improvements may appear moderate in absolute terms on some datasets, they are nevertheless significant when compared to prior works. As shown in the *Table 4*, CMNeXt improves over the RGB-only baseline by only 0.3 and 0.8 mIoU on RGB-Event and RGB-Lidar, respectively, while CMX even underperforms the unimodal baseline by 0.7 and 0.8 mIoU. These results highlight the inherent challenges in effective multimodal fusion. In contrast, our method achieves consistent gains of 0.8 and 1.9 mIoU over the RGB-only baseline, outperforming CMNeXt despite using **only 5.7M trainable parameters—less than 9.7% of those used in previous methods**. These results are especially encouraging, suggesting that our approach offers a more efficient and effective solution for resource-constrained scenarios.
>
>   *Table 4: Comparison of RGB-Event and RGB-Lidar segmentation performance on the DELIVER dataset.*
>   |Method|Modal|Trainable Params|mIoU|
>   |:-:|:-:|:-:|:-:|
>   |Segformer|RGB|25.8M|57.2|
>   |CMX|RGB-E|66.6M|56.5 (-0.7)|
>   |CMNeXt|RGB-E|58.7M|57.5 (+0.3)|
>   |**Ours**|**RGB-E**|**5.7M**|**58.0 (+0.8)**|
>   |CMX|RGB-L|66.6M|56.4 (-0.8)|
>   |CMNeXt|RGB-L|58.7M|58.0 (+0.8)|
>   |**Ours**|**RGB-L**|**5.7M**|**59.1 (+1.9)**|
>
> - Regarding Figure 5, we acknowledge that the overall scene-level outputs may appear similar at first glance. However, a closer inspection of the highlighted regions—already annotated in our submission version—reveals clear qualitative improvements over existing multimodal methods such as CMX and CMNeXt. Specifically, our method shows more accurate predictions in distant regions and for small, fine-grained structures, where prior approaches often produce fragmented or overly smoothed outputs.
> - Overall, we believe these quantitative and qualitative results jointly support the practical utility of MixPrompt, especially when considering the favorable trade-off between performance and parameter efficiency.

---

> > ### Comment · Reviewer_cD6A · 2025-08-05
> > **Official Comment by Reviewer cD6A**
> >
> > Thank you for the thoughtful and detailed rebuttal. I appreciate the additional experiments and clarifications, which satisfactorily address the specific questions I raised. I recognize the practical value of your lightweight and flexible design.
> >
> > That said, my original concern regarding the novelty of the approach remains. While the framework is effective and broadly applicable, the conceptual differences from prior prompting-based methods, particularly DPLNet, still appear incremental rather than fundamentally innovative.
> >
> > I hope the feedback is useful in further positioning and refining this promising direction.

---

> ### Author Response · Authors · 2025-08-05
> **Hope to Get Your Reply**
>
> Dear Reviewer cD6A,
>
> As the deadline is nearing, we wanted to gently follow up on our recent submission. We have meticulously addressed each of your suggestions one by one and incorporated these feedbacks into the revised version. During the rebuttal period, we thoroughly discussed the differences between MixPrompt and prior methods in terms of efficiency, robustness, and generalizability. Meanwhile, we **performed additional ablation studies on the hyperparameters of concern**, confirming the generality of our chosen settings.  Additionally, we **supplemented our response with theoretical analysis and experimental evidence to further validate the effectiveness of the multi-subspace design**. We hope that these efforts will alleviate your concerns regarding Mixprompt. Your feedback is highly valuable to us, and we would appreciate any updates or further guidance you might have regarding our revisions and responses.
>
> Thank you for your time and consideration.

---

> ### Comment · Area_Chair_bevN · 2025-08-05
> **Update after rebuttal**
>
> Dear Reviewer,
>
> The authors’ rebuttal has been posted. Please check the authors’ feedback, evaluate how it addresses the concerns you raised, and post any follow-up questions to engage the authors in a discussion. Please do this ASAP.
>
> Thanks.
>
> AC

---

> ### Author Response · Authors · 2025-08-07
> **(Round 2) Response to Reviewer cD6A**
>
> Dear Reviewer cD6A,
>
> We greatly appreciate your encouragement regarding Mixprompt, and it has been highly encouraging for us to address your concerns in the rebuttal. **In our initial rebuttal, we addressed your concerns regarding the differences between our approach and prior prompting-based methods, particularly DPLNet, focusing on efficiency, robustness, and generalizability**. **We sincerely appreciate your recognition of the practical value of our lightweight and flexible design**. We understand that your remaining concern may pertain to the methodological novelty of our work, and we would like to clarify the key distinctions once again.
>
> **Fundamental Design Differences**. DPLNet integrates learnable prompts directly within the attention modules of each backbone block, requiring architectural modifications and freezing modality-specific branches. This results in a more complex design with limited generalizability. In contrast, MixPrompt introduces modular prompts externally to the backbone, without altering its structure or needing modality-specific branches, offering a lighter and more flexible solution.
>
> **Multi-Subspace Prompting Strategy**. Our approach uniquely decomposes the feature space into multiple subspaces to finely align auxiliary modality information. This enables the seamless addition of new modalities (e.g., radar, event data) without retraining—an advantage not present in DPLNet.
>
> In summary, MixPrompt represents more than an incremental improvement and it embodies a paradigm shift from internal architecture modification to external modular guidance, resulting in an efficient, robust, and broadly generalizable multimodal fusion framework.
>
> We hope this clarifies the novelty of our work and kindly ask for your reconsideration of our assessment. Please feel free to let us know if you have any further questions or suggestions.
>
> With best regards!

---

### Official Review · Reviewer_DbRo · 2025-07-02

**Clarity:** 4
**Significance:** 3
**Originality:** 3
**Rating:** 5
**Confidence:** 4

**Summary:**

The paper introduces MixPrompt, an efficient method for integrating auxiliary modalities with RGB images for semantic segmentation through prompt tuning. MixPrompt utilizes a frozen backbone that extracts features exclusively from the RGB input. At each stage of the backbone, a lightweight prompter module is employed to effectively fuse RGB features with those from the auxiliary modality.
A key insight of the paper lies in the initialization of the prompt derived from the auxiliary modality. This initial prompt is merged with the RGB features *before* the first backbone block. To ensure a consistent alignment, the prompt is generated by processing the auxiliary input through the early layers of a pretrained RGB model.
The proposed method is evaluated on four datasets: NYU Depth V2, SUN-RGBD, MFNet, and DELIVER. MixPrompt consistently outperforms prior approaches while maintaining a significantly smaller parameter count.

**Questions:**

See the questions in the Strengths And Weaknesses section.

**Ethical Concerns:**

["NO or VERY MINOR ethics concerns only"]

**Final Justification:**

After the rebuttal, all of my original concerns were successfully addressed, so I decided to increase my rating.

**Limitations:**

yes

**Quality:**

3

**Strengths And Weaknesses:**

**Strengths**

* The paper introduces a **novel and efficient fusion strategy** for combining RGB images with auxiliary modalities (e.g., depth, thermal, event, LiDAR), requiring only a minimal number of learnable parameters.

* The design of the **initial prompt generation** mechanism offers an elegant and principled approach to the early fusion of modalities, ensuring meaningful feature alignment from the start of the network.

* The proposed method achieves **competitive or superior performance** on multiple benchmarks while maintaining a very small parameter footprint.

* The paper is **well written, clear, and easy to follow**, which facilitates understanding of the core ideas and implementation details.

---

**Weaknesses**

* **Related Works Section.** While the related work is included in the supplementary material, placing it outside the main paper is unfair in comparison to other submissions that incorporate it within the main text, often sacrificing space for methodology and experiments. I recommend including at least a condensed version of the related work section in the main paper to ensure a fair and comprehensive presentation.

* **Merging Strategy in the Mixed Prompting Module.** The two prompts ($h\_{i-1}$ and $e\_{i-1}$) are currently merged via simple summation. Have alternative merging strategies, such as concatenation, learnable weighted summation, attention-based fusion, or other parameter-efficient mechanisms, been explored? Investigating or at least discussing these alternatives could strengthen the contribution and potential generalizability of the proposed module.

---

> ### Author Rebuttal · Authors · 2025-07-27
>
> **We sincerely appreciate reviewer DbRo recognition of the novelty and efficiency of our proposed fusion strategy, elegant prompt generation design, and strong performance with minimal parameters**. We provide our feedbacks as follows regarding your valuable suggestions.
>
> ------
>
> > **Q1: Related Works Section. While the related work is included in the supplementary material, placing it outside the main paper is unfair in comparison to other submissions that incorporate it within the main text, often sacrificing space for methodology and experiments. I recommend including at least a condensed version of the related work section in the main paper to ensure a fair and comprehensive presentation.**
>
> - We agree with your point regarding fairness and presentation. **To address this concern, we promise to include a condensed version of the related works section in the main body of the paper in the camera-ready version, while retaining the full, detailed version in the supplementary material**. This revision will not affect other parts of the main text and help ensure a fair and comprehensive presentation of our work.
> - Specifically, condensed version will briefly cover two parts.
>   - **Semantic segmentation**. It summarizes the motivation for introducing multimodal inputs (e.g., depth, thermal) to address the limitations of RGB-only methods, reviews the two main existing research directions aboat multimodal semantic segmentation, and concludes the current challenges in this field.
>   - **Multimodal prompting**. It outlines how visual prompts, together with textual prompts, enhance cross-modal reasoning in vision-language models, and notes the limitations of early applications in semantic segmentation.
>
> ------
>
> > **Q2: Merging Strategy in the Mixed Prompting Module. The two prompts ($h\_{i-1}$ and $e\_{i-1}$) are currently merged via simple summation. Have alternative merging strategies, such as concatenation, learnable weighted summation, attention-based fusion, or other parameter-efficient mechanisms, been explored? Investigating or at least discussing these alternatives could strengthen the contribution and potential generalizability of the proposed module.**
> - We adopt a simple summation of the two prompt embeddings ($h\_{i-1}$ and $e\_{i-1}$), primarily due to its parameter-efficiency and empirical effectiveness in our preliminary experiments. While our current approach is effective, we fully recognize the potential of alternative fusion strategies—such as concatenation followed by a projection layer, learnable weighted summation via gating, or attention-based fusion—to enable more expressive cross-modal interactions, as the reviewer rightly pointed out. However, these methods typically introduce additional parameters and computational overhead, potentially undermining the objective of preserving a compact and efficient prompting module, while also increasing the complexity of training.
> - To futher investigate the effectiveness of these fusion methods, we conducted ablation studies on different prompt fusion strategies using the NYUDepth dataset. The quantitative results are shown in the table below, and the implementation details of each variant are described as follows.
>   - **Concatenation.** Concatenate $h\_{i-1}$ and $e\_{i-1}$ along the module dimension, and apply a linear transformation to fuse the combined representation back into the original embedding space.
>   - **Learnable Weighted Summation.** Introduce a learnable gating vector $g \in \mathbb{R}^C$ and compute the fused prompt as $\sigma(g) \cdot h\_{i-1} + (1 - \sigma(g)) \cdot e\_{i-1}$, where $\sigma$, where $\sigma$ denotes the sigmoid activation function. The gate $g$ is initialized as a trainable parameter and applied channel-wise to adaptively modulate the contribution of each modality.
>   - **Attention Fusion.** The fused prompt is obtained by computing cross-modal attention between $e\_{i-1}$ and $h\_{i-1}$: $\mathrm{Attn}(Q,K,V)=\mathrm{softmax}\left(\frac{QK\^{\intercal}}{\sqrt{d}}\right)V$.
>
>   *Table 1: Comparison of different prompt fusion strategies in the Mixed Prompting Module*
>
>   |Fusion Method|Trainable Params|FLOPs|mIoU (%)|
>   |:-:|:-:|:-:|:-:|
>   |Summation|5.74M| 109.01G |**60.1**|
>   |Concatenation|5.83M| 109.09G|59.4|
>   |Learnable Weighted Sum|5.74M|109.01G|59.5|
>   |Attention Fusion|5.84M|109.09G|57.0|
>
> - Overall, our results on NYUDepth indicate that simple summation achieves the best performance among the evaluated strategies, highlighting its effectiveness as a lightweight and robust fusion mechanism. Although concatenation and learnable weighted summation offer more flexible interactions between the two prompt streams, they do not yield noticeable performance improvements in our setting. One possible explanation is that the two prompts already provide sufficiently complementary information, and more complex merging may introduce redundancy or disrupt this balance. In contrast, attention fusion not only degrades performance, but also incurs substantial memory overhead. This is primarily due to the pairwise similarity computation between all token positions, which generates a large $N \times N$ attention map, where $N$ corresponds to the number of spatial positions in the feature map. For high-resolution inputs, $N$ can be large, making this step particularly memory-intensive.
> - These findings collectively support our use of simple summation as a practical and efficient fusion strategy in the mixed prompting module. We appreciate your suggestion and believe these findings provide a useful reference for future work exploring more adaptive fusion mechanisms in multimodal prompting.

---

> > ### Comment · Reviewer_DbRo · 2025-08-05
> > **Thanks for the rebuttal**
> >
> > Dear Authors,
> > Thank you for addressing my feedback and concerns. I found your study on the different merging strategies quite interesting, and I believe it would be valuable to include it in the supplementary material of your paper for future reference.
> >
> > Thank you for your time, and best of luck with your work!

---

> > > ### Comment · Area_Chair_bevN · 2025-08-08
> > > **Mandatory acknowledgement**
> > >
> > > Dear Reviewer,
> > >
> > > Could you please provide your mandatory acknowledgement for authors' rebuttal?
> > >
> > > Thank you.
> > >
> > > AC

---

> ### Author Response · Authors · 2025-08-05
> **Hope to Get Your Reply**
>
> Dear Reviewer DbRo,
>
> As the deadline approaches, we wanted to kindly follow up on our recent submission. We have carefully addressed each of your suggestions and incorporated the feedback into the revised version. During the rebuttal period, we conducted **additional ablation experiments on different prompt fusion strategies**. We have also discussed in detail the contents of the condensed version of the abstract. Your feedback has been invaluable, and we would greatly appreciate any updates or further guidance you may have regarding our revisions and responses.
>
> Thank you for your time and consideration.

---

> ### Comment · Area_Chair_bevN · 2025-08-05
> **Update after rebuttal**
>
> Dear Reviewer,
>
> The authors’ rebuttal has been posted. Please check the authors’ feedback, evaluate how it addresses the concerns you raised, and post any follow-up questions to engage the authors in a discussion. Please do this ASAP.
>
> Thanks.
>
> AC

---

> ### Author Response · Authors · 2025-08-05
> **Thank you to your recognition and encouragement!**
>
> Dear Reviewer DbRo,
>
> Thank you very much for your recognition. We are greatly encouraged by the opportunity to address your concerns. The study on  different merging strategies will be included in the final version of the paper. Upon acceptance, we will publicly release the corresponding code.
>
> Once again, we sincerely appreciate your recognition.
>
> With best regards!

---

### Comment · Area_Chair_bevN · 2025-08-01
**Authors' rebuttal posted and discussion**

Dear Reviewers,

Thank you for your efforts in reviewing this paper. The authors' rebuttal has been posted. This paper received diverse initial ratings. Please read the rebuttal materials and comments from other reviewers to justify if your concerns have been resolved and update your final rating with justifications.

AC

---

### Note · Authors · 2025-08-14

**Dear Reviewers, ACs, and SACs,**

We sincerely thank you for the positive recognition of our work, the constructive feedback, and the valuable time and effort you have devoted—especially the AC’s careful coordination throughout the review process.

MixPrompt targets a practical gap in multimodal semantic segmentation: attaining strong cross-modal gains with **minimal trainable parameters** and **no backbone surgery**. Our core contribution is lightweight prompting module and multi-subspace prompt mixing that decouples modality adaptation from a frozen RGB backbone, enabling a compact, modality-agnostic solution.

**What is fundamentally new vs. prior prompting (e.g., DPLNet/CBE/OmniVec):**

 • We do not inject token prompts into backbone attention nor build modality-specific branches; instead we project and mix prompts in multiple learnable subspaces external to the backbone, preserving efficiency and extensibility.

 • This design **generalizes across five modality pairs** (RGB-D/T/L/E) **without architecture changes**; in contrast, DPLNet shows modality sensitivity (underperforms CMNeXt on MFNet).

 • Unlike OmniVec-style large pretraining, MixPrompt achieves competitive accuracy with **5.7M trainable params** and **frozen** backbone.

**Key evidence supporting the design:**

 • **Efficiency:** 5.7M trainable params (<10% of many fusion baselines) yet SOTA or competitive accuracy.

 • **Robustness:** On DELIVER and MFNet (night/fog/blur/exposure), MixPrompt consistently surpasses CMX/CMNeXt, where some fusion methods even degrade below RGB-only.

 • **Pretraining:** Re-training CMX and CMNeXt with ADE20K-pretrained weights yields only modest gains (+1.2–1.4 mIoU), while our method still surpasses them by 2.9–3.1 mIoU with just 5.7M parameters. These results validate both the fairness of our comparisons and the effectiveness of our approach.

**Camera-ready commitments:** we will (i) move a concise **Related Work** into the main paper (ii) add **trainable-parameter columns** and **RGB-only baselines** in the main tables, (iii) include the **above ablations** and **clear setting tables**, and (iv) provide **visual analyses** of subspace behavior.

We believe the clarifications and new evidence provided here address all concerns and demonstrate the technical value and practical impact of our work, and we look forward to your positive consideration.

With best regards!

---

### Decision · Program_Chairs · 2025-09-17

**Decision:**

Accept (poster)

**Comment:**

This paper proposes a novel efficient mixed prompting method for multimodal semantic segmentation. The authors conduct experiments on multiple benchmarks to validate the effectiveness of the proposed method. Initially, the reviewers are concerned on the motivation, method details, and experiments of this work. The authors provide an effective rebuttal that addresses most of the questions. The AC agrees with most reviewers that this work provides some interesting insights and design for efficient multimodal semantic segmentation. Considering all factors, the AC recommends accepting this paper. The authors should integrate their responses in the final version.